# `OmniVideo-R1`: Reinforcing Audio-visual Reasoning with Query Intention and Modality Attention

**Zhangquan Chen**[* 1]   **Jiale Tao**[† 2]   **Ruihuang Li**[2]   **Yihao Hu**[3]   **Ruitao Chen**[2]   **Zhantao Yang**[2]   **Xinlei Yu**[4]   **Haodong Jing**[5]   **Manyuan Zhang**[6]   **Shuai Shao**[2]   **Biao Wang**[2]   **Qinglin Lu**[2]   **Ruqi Huang**[† 1]

## Abstract

While humans perceive the world through diverse modalities that operate synergistically to support a holistic understanding of their surroundings, existing omnivideo models still face substantial challenges on audio-visual understanding tasks. In this paper, we propose OMNIVIDEO-R1, a novel reinforced framework that improves mixed-modality reasoning. OMNIVIDEO-R1 empowers models to "think with omnimodal cues" by two key strategies: (1) query-intensive grounding based on self-supervised learning paradigms; and (2) modality-attentive fusion built upon contrastive learning paradigms. Extensive experiments on multiple benchmarks demonstrate that OMNIVIDEO-R1 consistently outperforms strong baselines, highlighting its effectiveness and robust generalization capabilities. Our code is available at https://github.com/zhangquanchen/OmniVideo-R1.

## 1. Introduction

Human cognition is inherently multimodal; we perceive the physical world by processing visual and auditory signals in parallel, integrating them to construct a coherent understanding of complex environments (Zhou et al., 2025c; Benchekroun et al., 2023; Li et al., 2025a). As Large Language Models (LLMs) evolve into Multimodal LLMs (MLLMs), the ability to interpret such multisensory inputs has become a cornerstone of artificial general intelligence. However, contrary to the expectation that more modalities yield better understanding, current omnimodal models often

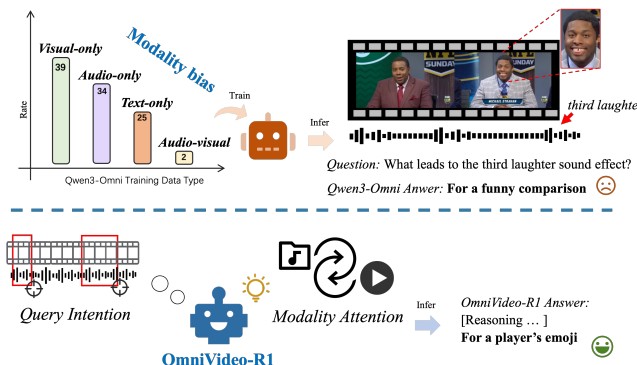

*Figure 1.* Pre-trained MLLMs (e.g., Qwen3-Omni) often exhibit suboptimal performance in audio-visual reasoning tasks due to inherent modality bias. To address this limitation, we reinforce the audio-visual reasoning ability by leveraging query intention and modality attention.

exhibit a paradoxical behavior.

This phenomenon is also evident in the state-of-the-art models. As shown in Fig. 1, pre-training inherently involves trade-offs across heterogeneous tasks, which can induce a natural modality bias. Consequently, within the Qwen3-30B-A3B family, the Omni variant (Xu et al., 2025c) (audio-visual) substantially underperforms the VL variant (Bai et al., 2025a) (visual-only), dropping from 72.1 to 68.5 on MMStar (Chen et al., 2024) and from 80.1 to 75.9 on MathVista_mini (Lu et al., 2023). These results expose a key limitation of current paradigms: instead of synergistic fusion, *incorporating the audio modality can undermine the model's established visual reasoning capability*.

A natural response is to increase mixed audio-visual supervision during pre-training; however, scaling high-quality mixed-modality data and aligning it with downstream reasoning needs is non-trivial. On the other hand, existing post-training pipelines commonly rely on supervised fine-tuning (SFT) or vanilla reinforcement learning (RL) (e.g., GRPO) (Zhao et al., 2025b; Xing et al., 2025; Yang et al., 2025b; Zhang et al., 2023; Sun et al., 2024). Yet these post-training methods do not explicitly train *audio-visual mixed-modality reasoning* behaviors, such as locating and composing evidence across modalities. That is, they provide

---

* The work was conducted during the internship of Zhangquan Chen (czq23@mails.tsinghua.edu.cn) at Tencent HY. [1]THU [2]Tencent HY [3]HNU [4]NUS [5]XJTU [6]CUHK. Correspondence to: Ruqi Huang <ruqihuang@sz.tsinghua.edu.cn>, Jiale Tao <jiale-tao.std@gmail.com>.

little supervision over intermediate evidence-tracking. As a result, the model may *ignore decisive audio or visual cues and still produce the correct answer by exploiting dataset biases or unimodal shortcuts*.

To address this challenge, we present OmniVideo-R1, the first post-training framework designed to improve mixed-modality reasoning. We posit that solving such problem requires more than just balancing datasets; it requires instilling robust reasoning behaviors that allow the model to actively select and fuse information. Specifically, OmniVideo-R1 optimizes two fundamental capabilities: (1) **query-intensive grounding** and (2) **modality-attentive fusion** built upon query-intensive reasoning.

We first introduce *query-intensive grounding*. Inspired by the "think with images" paradigm from OpenAI-o3 (OpenAI, 2025), we enable the model to explicitly locate and reason about specific audio-visual segments relevant to the user's query before generating a response. To achieve this without expensive process-level annotations, we design a *self-supervised training scheme* utilizing interleaved grounding and caption pairs. This mechanism allows the model to generate its own grounding hypotheses and validate them against textual descriptions.

However, grounding alone is not sufficient. In practice, models trained only with query-intensive behaviors may still under-utilize certain key audio evidence (e.g., the subtle but decisive sound cues in Fig. 5), resulting in biased evidence usage. Therefore, we propose *modality-attentive fusion* to further strengthen deep audio-visual integration. We design a *contrastive learning-based strategy* that explicitly encourages the model to derive higher confidence from mixed audio-visual inputs compared to single-modality counterparts. This forces the model to discover synergistic relationships between visual events and auditory cues, ensuring that the fused representation is strictly superior to its constituent parts.

By combining these strategies into a unified RL framework, OmniVideo-R1 turns mixed-modality understanding into a query-driven reasoning process with audio-visual cues.

Our primary contributions are summarized as follows:

- We propose OmniVideo-R1, the first RL-based framework designed to improve mixed-modality reasoning.

- We construct a high-quality corpus of 88K audio–visual training samples through a dedicated data-cleaning pipeline, specifically curated to support complex reasoning tasks.

- We introduce a two-stage RL paradigm that incorporates *self-supervised grounding* and *contrastive fusion*, enabling the model to learn query intention and modal-

ity attention without relying on process-level annotations.

- Extensive experiments demonstrate that OmniVideo-R1 consistently outperforms strong baselines on audio-visual benchmarks while effectively maintaining robust visual-only performance.

## 2. Related Work

### 2.1. Omnimodal Large Language Models

The integration of audio and visual modalities is closer to real-world recordings, and requires models to form a cohesive understanding of the surroundings, like humans (Zhao et al., 2025b). Early efforts always focused on silent video understanding (Bai et al., 2025b; Zhang et al., 2024; Feng et al., 2025a; Chen et al., 2025c) or treated audio a simple add-on to text (Li et al., 2025b), which fragments natural omnimodal representations and thereby limits performance.

Consequently, subsequent works have aimed for deeper cross-modal fusion. MiniCPM-o-2.6 (Yao et al., 2024) and Baichuan-Omni-1.5 (Li et al., 2024b) extend vision–language foundations with audio processing capabilities, enabling operation across a broader range of modalities. Ola (Liu et al., 2025) adopts a progressive modality-alignment strategy that incrementally strengthens the language model's ability to exploit additional modalities. The Video-LLaMA series (Zhang et al., 2023) concatenates audio and visual tokens to support joint audio–video understanding, whereas the Video-SALMONN (Sun et al., 2024) series employs a multi-resolution causal Q-Former (Li et al., 2023) to process audio and video simultaneously. Moreover, InternVideo series (Wang et al., 2022) aligns video with audio events, speech, and text via cross-modal contrastive learning, thereby facilitating integrated audio–video representation learning. Qwen2.5-Omni (Xu et al., 2025b) introduces a "thinker–talker" architecture, an end-to-end multimodal framework capable of perceiving diverse input types. More recently, Qwen3-Omni (Xu et al., 2025c) leverages an Audio Transformer (AuT) for audio encoding and incorporates TM-RoPE, further enhancing the audio–visual understanding capabilities.

Despite these advances, current multimodal models *still exhibit substantial limitations on complex tasks*, particularly in scenarios that demand tightly *integrated audio–visual understanding* or more *sophisticated logical reasoning*.

### 2.2. Reinforced Multimodal Reasoning

Reinforcement learning has become a widely adopted paradigm for enhancing the performance of large language models (Zang et al., 2025; Zang, 2025). Recent work combines RL with vision and language to elicit stronger reason-

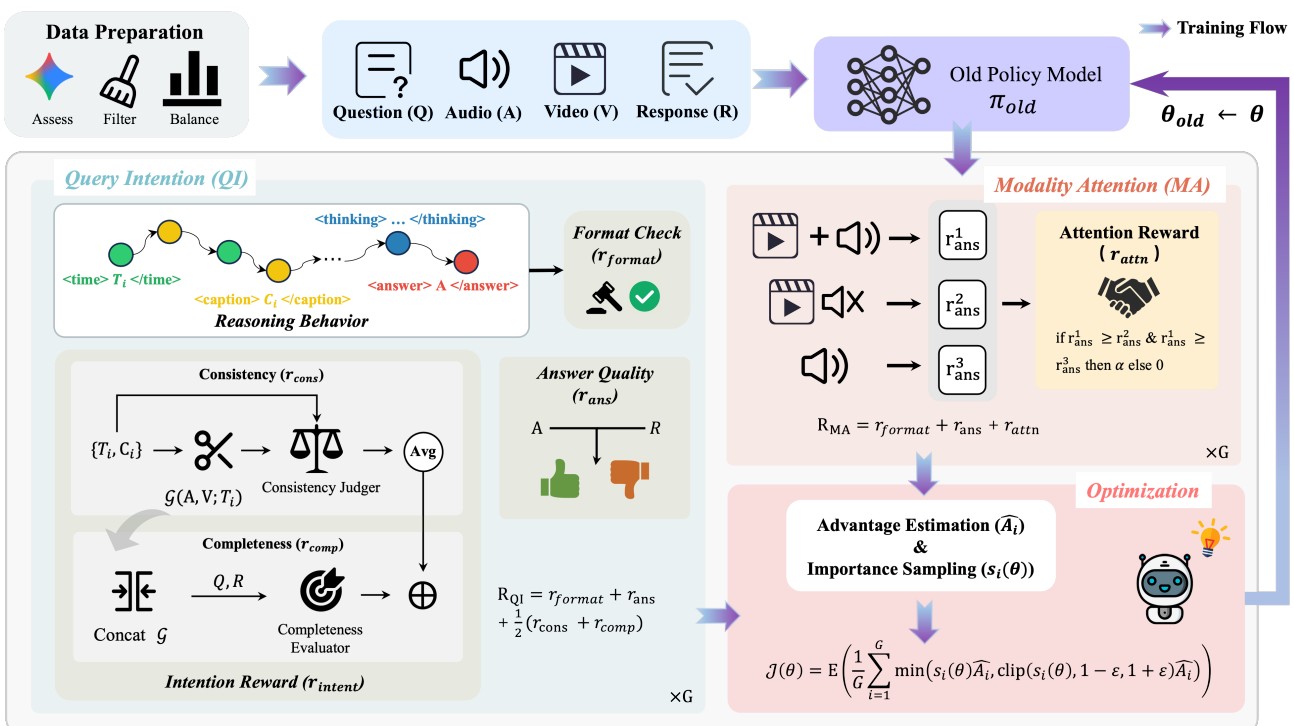

**Figure 2.** The schematic illustration of our OmniVideo-R1. Based on the dataset collected from data preparation, our training consists of two stages: (1) QI stage establishes query-intensive grounding behavior by aligning multiple time–caption pairs without process-level annotations. (2) MA stage further performs modality-attentive fusion by optimizing a contrastive modality reward.

ing capabilities. Some approaches, inspired by DeepSeek-R1 (Guo et al., 2025), introduce purely textual chain-of-thought (Thawakar et al., 2025; Chen et al., 2025a; Dong et al., 2024; Zang & Liu, 2024). Building on this, methods such as VisRL (Chen et al., 2025b), SIFThinker (Chen et al., 2025d), GRIT (Fan et al., 2025), and CogCoM (Qi et al., 2024) enable "thinking with images" by integrating visual evidence into the reasoning trajectory.

Beyond these efforts, several studies have extended the notion of reasoning to omnimodal models. R1-Omni (Zhao et al., 2025a) is primarily designed for audio–visual referring segmentation, whereas EchoInk-R1 (Xing et al., 2025) investigates the direct application of vanilla GRPO (Guo et al., 2025) in the omnimodal setting. In addition, Omni-R1 (Zhong et al., 2025) adopts a dual-system architecture to tackle long-horizon video–audio reasoning, and HumanOmnv2 (Yang et al., 2025b) enhances model capabilities through training on datasets curated for complex human intention understanding.

However, compared with silent video reasoning (Feng et al., 2025b; Wang et al., 2025a), *current explorations of omnimodal reasoning remain relatively limited*. Existing approaches concentrate on directly transferring vanilla RL, designing intricate multi-branch architectures, or constructing specialized training datasets. Yet omnimodal models *inher-*

*ently require deeper multimodal fusion in order to unlock stronger reasoning capabilities*, thereby achieve genuine "aha moments." Consequently, there is still a conspicuous absence of training methodologies that are tailored to the distinctive characteristics of such omnimodal models.

## 3. Methodology

**Preliminary.** Reinforcement learning (Christiano et al., 2017) has emerged as a particularly effective approach for substantially enhancing the robustness and factual accuracy of large language models (Ouyang et al., 2022). In practice, off-policy learning settings are typically used during policy model training to improve sample efficiency. However, for Mixture-of-Experts (MoE) models (e.g., Qwen3-Omni-30B-A3B (Xu et al., 2025c)), the activation of different experts can induce substantial shifts in the token distribution. Under such conditions, token-level importance sampling often introduces high-variance noise into the training gradients, which accumulates over long sequences and is further exacerbated by clipping mechanisms. To this end, our method performs optimization directly at the sequence level, following the formulation introduced by Group Sequence Policy Optimization (GSPO) algorithm (Zheng et al., 2025). The optimization objective is formulated as:

$$\mathcal{J}(\theta) = \mathbb{E}_{x\sim\mathcal{D},\{y_i\}_{i=1}^{G}\sim\pi_{\theta_{\text{old}}}(\cdot|x)}\left(P_\theta\right), \qquad (1)$$

where the response $y_i$ is sampled from old policy model $\pi_{\theta_{\text{old}}}$ based on the input $x$, and $P_\theta$ is:

$$P_\theta = \frac{1}{G}\sum_{i=1}^{G}\min\left(s_i(\theta)\widehat{A}_i, \text{clip}\left(s_i(\theta), 1-\varepsilon, 1+\varepsilon\right)\widehat{A}_i\right). \qquad (2)$$

Here, we also adopt the group-based advantage estimation:

$$\widehat{A}_i = \frac{R(x,y_i) - \text{mean}\left(\{R(x,y_i)\}_{i=1}^{G}\right)}{\text{std}\left(\{R(x,y_i)\}_{i=1}^{G}\right)}, \qquad (3)$$

$R(\cdot)$ denotes the reward function that will be introduced below, and we define the importance ratio $s_i(\theta)$ based on sequence likelihood (Zheng et al., 2023):

$$s_i(\theta) = \exp\left(\frac{1}{|y_i|}\sum_{t=1}^{|y_i|}\log\frac{\pi_\theta(y_{i,t}|x, y_{i,<t})}{\pi_{\theta_{\text{old}}}(y_{i,t}|x, y_{i,<t})}\right). \qquad (4)$$

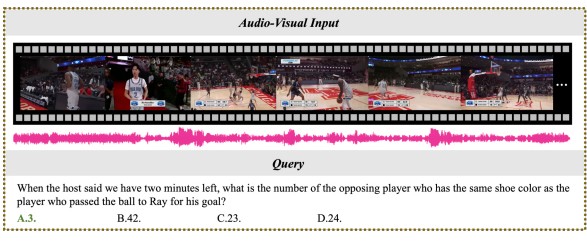

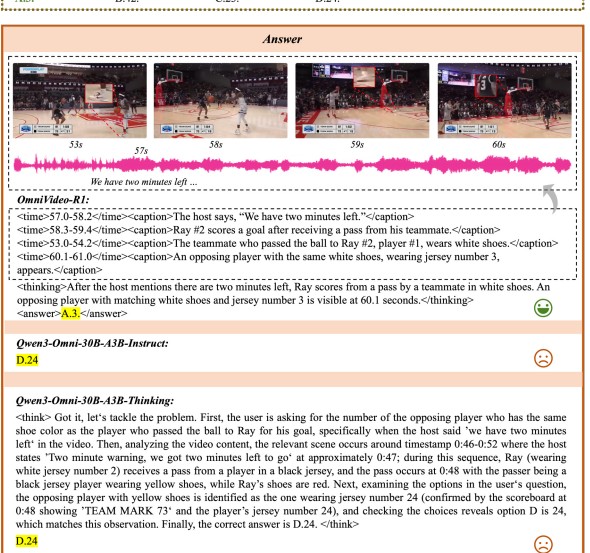

*Figure 3.* Visualization of the responses and underlying reasoning process generated by OmniVideo-R1 and Qwen3-Omni-30B-A3B-Instruct, -Thinking to an audio-visual understanding question.

**Method overview.** As shown in Fig. 2, OmniVideo-R1 adopts GSPO to optimize the entire reasoning process, enabling the model to extract intention-relevant cues and

to effectively integrate audio–visual information throughout reasoning. This model behavior emerges through two training stages. That is, we first induce the model to develop query-intensive reasoning behavior, and then, further train it to integrate multiple modalities in a logically consistent manner. In the first stage (QI), the model is trained with a *self-supervised objective* defined over multiple pairs of grounding and caption generated within the reasoning trajectory. In the second stage (MA), we promote deeply fused understanding by first decoupling the modality-specific inputs and then performing *contrastive learning* across them. Notably, throughout the entire training pipeline, OmniVideo-R1 doesn't rely on any explicit process-level annotations for query-intensive grounding or modality fusion.

Fig. 3 illustrates our reasoning process in comparison with Qwen3-Omni-30B-A3B. Our OmniVideo-R1 endows the model with the ability to "think with omnimodal cues", i.e., to perform query-intensive grounding that identifies key cues, thereby enabling more accurate and reliable reasoning to the final answer.

### 3.1. Data Preparation

We first collect raw data from LLaVA-Video (Zhang et al., 2024) and Video-Vista (Li et al., 2024a), and perform structural validation to remove problematic samples with metadata issues (e.g., silent videos). To further filter out low-quality samples that are misaligned with our multimodal setting, we apply a three-stage refinement pipeline as shown in Fig. 4, which consists of (i) quality assessment, (ii) heuristic filtering, and (iii) categorical balancing.

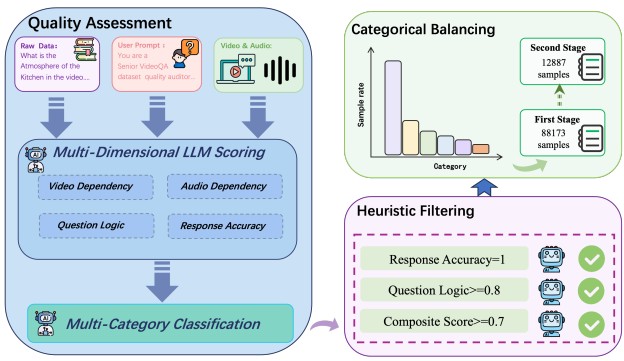

*Figure 4.* Pipeline for our data preparation consisting of 3 stages.

For data quality assessment, we employ Gemini-2.5-Pro (Google, 2024) to score each sample along four key dimensions: video dependency $s_v$, audio dependency $s_a$, question logic $s_q$, and response accuracy $s_r$. Each dimension is normalized to a maximum of 1, and the weighted composite score $s_c$ is computed as a weighted sum over the four dimensions. Subsequently, Qwen-3-32B (Yang et al.,

2025a) is used to categorize the samples according to the 15-category taxonomy described in the Appendix.

After scoring and categorizing data, we apply the following filtering rules: (i) $s_r = 1$, (ii) $s_q \geq 0.8$, (iii) $s_c \geq 0.7$. Any sample that fails to satisfy any of these rules is discarded.

Finally, we mitigate long-tail bias by pruning sparse categories (i.e., those with fewer than 10 samples). Observing a significant gap between the top two classes, we further require that the number of samples in the largest class does not exceed three times that of the second-largest class. Specifically, we first retain all samples with both $s_a = 1$ and $s_i = 1$, then sort the remaining data in descending order of $s_c$ and remove samples that exceed the specified count, resulting in a smoother data distribution.

After applying all of the above steps, we obtain 88173 examples for the first training stage training. Considering the high-quality requirements for audio-visual fused data in the second training stage, we then derive a subset of 12887 examples by keeping only instances with high audio-visual dependency, i.e., $s_v \geq 0.7$ and $s_a \geq 0.7$.

### 3.2. Query-intensive Grounding (QI)

Query-intensive grounding operations aim to help the model identify key frames containing critical audio-visual cues within a video sequence (Wang et al., 2025a). However, human annotations of prompt-related key frames are often complex and expensive. To address this issue, we propose a novel grounding approach that establishes a correspondence between grounding and captioning *without relying on any dense annotations*, thereby enabling *self-supervised learning of the model's procedural behavior*.

Specifically, given one question and the corresponding audio–visual content $(Q, A, V)$, we encourage the model to produce outputs in the structured format `<time>...</time><caption>...</caption> ... <thinking>...</thinking><answer>...</answer>`. A reward of $r_{\text{format}} = 1.0$ is assigned to responses that strictly comply with this output template. For each rollout, we denote the multiple generated time–caption pairs by $\{T_1, C_1, T_2, C_2, \ldots, T_N, C_N\}$. We then perform self-supervised learning by evaluating the consistency reward between each $T_i$ and $C_i$, i.e.,

$$r_{\text{cons}} = \frac{1}{N} \sum_{i=1}^{N} E_{\text{cons}}^{(L)}\big(\mathcal{G}(A, V; T_i), C_i\big), \qquad (5)$$

where $\mathcal{G}(A, V; T_i)$ extracts the audio-visual segment from $(A, V)$ corresponding to the time span $T_i$, and $E_{\text{cons}}^{(L)}(\cdot)$ denotes a soft evaluation function implemented via a judger model (i.e., Qwen3-VL-235B-A22B-Instruct) with $L$ predefined rules. The detailed rules and the associated prompts are provided in Appendix.

On the one hand, we perform self-supervised learning by enforcing the correctness of each time–caption pair. On the other hand, we also require the grounding to be precise, i.e., it should (i) minimally and effectively cover all ground-truth intention-related cues $T_{\text{gt}}$, and (ii) avoid redundant predictions. Formally, for each $i, j \leq N$, we xpect:

$$\left[\left(\bigcup_{i=1}^{N} T_i\right) \cap T_{\text{gt}} = T_{\text{gt}}\right] \wedge \left[T_i \cap T_j = \varnothing, \, \forall\, i \neq j\right]. \quad (6)$$

However, in this work, we tackle the challenging setting where no ground-truth $T_{\text{gt}}$ is available, and instead propose a soft approximation to solve Eq. 6. Specifically, we first crop all predicted segments and then concatenate them into a single sequence, which is subsequently evaluated along two dimensions: content completeness and precision. In other words, we assess whether the audio-visual information contained within the grounded segments is *adequate and accurate for supporting the reasoning process from the question Q to the final answer R*. Accordingly, we define the completeness reward as:

$$r_{\text{comp}} = E_{\text{comp}}^{(M)}\Big(\bigoplus_{i=1}^{N} \mathcal{G}(A, V; T_i), \, Q, \, R\Big), \qquad (7)$$

where $\bigoplus_{i=1}^{N} \mathcal{G}(A, V; T_i)$ denotes the temporally ordered concatenation of all grounded audio-visual segments, yielding a single composite video clip. Here, $E_{\text{comp}}^{(M)}(\cdot)$ is the intent evaluation function instantiated with $M = 3$ predefined rules. More details are listed in the Appendix.

Meanwhile, we also leverage the outcome signal, following (Guo et al., 2025). Specifically, we softly evaluate the quality of the final answer and assign a continuous score $r_{\text{ans}}$; the detailed evaluation protocol is provided in Appendix. Finally, the reward in our QI training stage is defined as

$$R_{\text{QI}} = r_{\text{format}} + r_{\text{ans}} + \frac{1}{2}\big(r_{\text{cons}} + r_{\text{comp}}\big). \qquad (8)$$

We establish a unified training framework from three complementary perspectives: (i) global format regularization $r_{\text{format}}$, (ii) outcome-based constraints $r_{\text{ans}}$, and (iii) process-level self-supervision $r_{\text{intent}} = \frac{1}{2}(r_{\text{cons}} + r_{\text{comp}})$. Under this training design, the model is expected to *infer the underlying intention, extract task-relevant cues, and perform reasoning over these audio–visual content*.

### 3.3. Modality-attentive Fusion (MA)

As QI stage is primarily evaluated in a vision-centric manner, relying solely on query-intensive grounding still prevents the model from capturing the subtle but decisive sound cues (as shown in Fig. 5). This inability to leverage audio cues

| Method | Daily-Omni | WorldSense | IntentBench | VideoHolmes |
|---|---|---|---|---|
| *Closed-source Models* | | | | |
| Gemini-2.0-Flash | 67.8 | 56.2 | 67.8 | 30.6 |
| Gemini-2.5-Pro (Google, 2024) | 81.4 | 64.6 | 67.2 | 64.9 |
| Gemini-3-Pro | 81.1 | 66.4 | 71.5 | 67.0 |
| *Open-source Models* | | | | |
| VideoLLaMA2-7B (Cheng et al., 2024) | 35.2 | 25.4 | — | 35.2 |
| Qwen2.5-Omni-7B (Xu et al., 2025a) | 47.5 | 45.4 | 64.2 | 16.4 |
| MiniCPM-o-7B (Yao et al., 2024) | 53.1 | — | 54.5 | — |
| VITA-1.5-7B (Fu et al., 2025b) | — | 36.9 | 54.2 | — |
| Ola-7B (Liu et al., 2025) | 49.9 | — | 57.4 | — |
| HumanOmniV2-7B (Yang et al., 2025b) | 58.5 | 47.1 | 69.3 | — |
| video-SALMONN 2+-7B (Tang et al., 2025) | 71.8 | 50.9 | — | 46.9 |
| video-SALMONN 2+-72B (Tang et al., 2025) | 79.4 | 56.5 | — | 57.8 |
| Qwen3-Omni-30B-A3B-Instruct (Xu et al., 2025c) | 63.6 | 54.0 | 65.7 | 49.0 |
| Qwen3-Omni-30B-A3B-Thinking (Xu et al., 2025c) | 75.8 | 48.0 | 68.5 | 57.3 |
| OmniVideo-R1 | **82.8** | **65.8** | **74.2** | **62.9** |

*Table 1.* Performance of different methods on a range of audio-visual benchmarks, including Daily-Omni (Zhou et al., 2025b), WorldSense (Benchekroun et al., 2023), IntentBench (Yang et al., 2025b), and VideoHolmes (Cheng et al., 2025). Our training was conducted on QI and on both QI + MA. The **best** is highlighted, and the second-best is underlined.

further leads to substantial redundant outputs (as shown of Fig. 12 in the Appendix). To address this issue, we propose a *modality-attentive fusion* scheme, whose central idea is to encourage the model to *fully exploit and synergistically integrate both audio and visual information to improve accuracy*.

Concretely, for each input $x$, we compare the model's performance under three rollout settings: (i) combined audio–visual input; (ii) silent-video-only input; and (iii) audio-only input. For a desirable multimodal understanding model, the performance with full multimodal input should not be inferior to that with any single-modality input, especially on datasets where both acoustic and visual cues are required to correctly answer the question. Denote the soft scores associated with these three rollouts by $r_{\text{ans}}^1$, $r_{\text{ans}}^2$, and $r_{\text{ans}}^3$, respectively. We then define the *attention* reward as

$$r_{\text{attn}} = \begin{cases} \alpha, & \text{if } r_{\text{ans}}^1 \geq r_{\text{ans}}^2 \text{ and } r_{\text{ans}}^1 \geq r_{\text{ans}}^3 \\ 0, & \text{otherwise} \end{cases} \quad (9)$$

where $\alpha$ is a hyperparameter controlling the magnitude of the *attention* reward (set to $\alpha = 0.3$ in our experiments). This contrastive formulation explicitly encourages the model to achieve superior performance when audio and visual information are effectively fused, rather than relying predominantly on a single modality.

Building upon the *contrastive learning* strategy, our MA training stage focuses on enhancing model capabilities in a more targeted subset of data which specifically requires integrated audio–visual understanding. This stage aims to advance the reasoning paradigm from query-intensive grounding to deeper multimodal understanding. Formally, the reward for this stage is defined as:

$$R_{\text{MA}} = r_{\text{format}} + r_{\text{ans}} + r_{\text{attn}}. \quad (10)$$

## 4. Experiments

We evaluate OmniVideo-R1 with several state-of-the-art (SOTA) methods on an array of categories as follows. More details about benchmarks are listed in the Appendix.

**Training.** OmniVideo-R1 is trained based on Qwen3-Omni-30B-A3B following the pipeline described in Sec. 3. As detailed in Sec. 3.1, we use 88173 samples for QI stage (Sec. 3.2) and 12887 samples for MA stage (Sec. 3.3).

**Hyper-parameters.** For OmniVideo-R1, we conduct training under a 128×H20 setup with a global batch size of 256. We set the balancing coefficient of all rewards to 1 and use a learning rate of $1 \times 10^{-6}$. The rollout number is 8, and the maximum sequence length is 32768. Additional details are provided in the Appendix.

**Evaluation metric.** For multiple-choice questions, we report Accuracy, which is calculated based on exact matches between the model's predictions and the ground truth.

### 4.1. Omnimodal Understanding

We first assess OmniVideo-R1 on a suite of audio-video understanding benchmarks as shown in Tab. 1. After the OmniVideo-R1 training phase, the model shows remarkable improvements across multiple benchmarks. Notably, OmniVideo-R1 outperforms the open-source SOTA model Video-SALMONN 2+-72B (which has a larger parameter scale) by at least **4.3%** (82.8 vs. 79.4). Additionally, on specific benchmarks, OmniVideo-R1 even exceeds the latest closed-source SOTA model Gemini3-Pro, achieving a **2.1%** advantage (82.8 vs. 81.1) on Daily-Omni and a **3.8%** improvement (74.2 vs. 71.5) on IntentBench.

Interestingly, certain reasoning-oriented variants have been

| Method | Audio Type | | | Video Duration | | | | Avg. |
|---|---|---|---|---|---|---|---|---|
| | Music | Sound | Speech | (0,1] min | (1,5] min | (5,10] min | (10,30] min | |
| *Closed-source Models* | | | | | | | | |
| Gemini-2.0-Flash | 29.7 | 40.3 | 43.2 | 49.4 | 43.2 | 41.1 | 34.9 | 41.5 |
| Gemini-2.5-Pro (Google, 2024) | 38.5 | 57.7 | 61.7 | 57.8 | 64.4 | 55.0 | 55.9 | 58.9 |
| Gemini-3-Pro | 56.2 | 54.1 | 55.7 | 61.0 | 56.4 | 52.9 | 52.5 | 55.5 |
| *Open-source Models* | | | | | | | | |
| VideoLLaMA2-7B (Cheng et al., 2024) | 26.4 | 30.7 | 29.3 | 32.0 | 28.2 | 29.6 | 28.3 | 29.2 |
| Qwen2.5-Omni-7B (Xu et al., 2025b) | 23.1 | 25.3 | 30.7 | 41.6 | 27.4 | 25.3 | 26.7 | 29.3 |
| MiniCPM-o-7B (Yao et al., 2024) | 27.5 | 28.6 | 30.2 | 31.4 | 28.5 | 34.5 | 26.2 | 29.7 |
| HumanOmniV2-7B (Yang et al., 2025b) | 20.9 | 31.1 | 31.6 | 36.6 | 29.4 | 29.6 | 29.3 | 30.5 |
| Baichuan-Omni-1.5-7B (Li et al., 2024b) | 24.2 | 31.3 | 31.4 | 28.9 | 31.8 | 28.4 | 32.4 | 30.7 |
| Qwen3-Omni-30B-A3B-Instruct (Xu et al., 2025c) | 30.8 | 35.8 | 38.0 | 46.8 | 37.4 | 37.7 | 29.7 | 37.0 |
| Qwen3-Omni-30B-A3B-Thinking (Xu et al., 2025c) | 26.4 | 37.2 | 38.5 | 46.8 | 35.6 | 35.5 | 35.2 | 37.2 |
| OmniVideo-R1 | **40.7** | **38.1** | **46.6** | **53.8** | **43.5** | **43.9** | **41.4** | **44.8** (+7.8pp) |

*Table 2.* Accuracy comparison of OmniVideo-R1 and other methods on OmniVideoBench (Li et al., 2025a). The **best** is highlighted and the second-best is underlined. The performance gains of our method over the base model are indicated in red parentheses.

observed to underperform compared to their base counterparts on specific benchmarks (e.g., Qwen3-Omni-30B-A3B-Thinking vs. Qwen3-Omni-30B-A3B shows 48.0 vs. 54.0 on WorldSense). In contrast, OmniVideo-R1 consistently demonstrates superior performance over the base model across all evaluated benchmarks, underscoring both the effectiveness and robustness of our approach.

Furthermore, we evaluate OmniVideo-R1 on a more challenging benchmark focused on synergistic audio-visual understanding, with a strong emphasis on modality complementarity and logical consistency. As shown in Tab. 2, OmniVideo-R1 surpasses Qwen3-Omni-30B-A3B by **21.1%** (44.8 vs. 37.0). Previous methods performed close to random guessing on this benchmark, but OmniVideo-R1 breaks through this bottleneck and achieves significant gains, consistently surpassing the base model across all evaluation dimensions. These results highlight the substantial potential of audio-visual joint reasoning through accurately grounded key cues.

### 4.2. Visual-only Understanding

On the other hand, to assess whether the model suffers performance degradation in a single modality after mixed-modality post-training, we evaluate OmniVideo-R1 on a suite of silent-video benchmarks. As shown in Tab. 3, OmniVideo-R1 exhibits no evident degradation and even demonstrates improvements compared to the base model; specifically, it achieves gains of **4.4%** (73.6 vs. 70.5), **-1.4%** (74.1 vs. 75.2), and **3.4%** (51.9 vs. 50.2) on Video-MME, MLVU, and LVBench, respectively.

This robustness stems from the model's ability to effectively ground behaviors during inference, allowing it to proficiently capture key cues regardless of whether the input is purely visual or audio-visual. These results confirm our core objective of fostering modality integration to enhance reasoning, rather than resulting in trade-offs between modalities.

### 4.3. Different Training Strategies

Following the dataset $\mathcal{D}$ curated for OmniVideo-R1, we first attempt to use these 88173 examples to directly learn the final response in the QA SFT setting, as reported in Tab. 4. That is, the model is supervised only on the final answers. In contrast, CoT SFT augments $\mathcal{D}$ with chain-of-thought annotations generated by Gemini-2.5- Pro (Google, 2024) and then fine-tunes the model on these CoT-augmented examples. Vanilla RL instead applies standard GRPO (Guo et al., 2025) on $\mathcal{D}$ under a `<think>...</think><answer>...</answer>` protocol, using a mixture of format and soft-response scores as the reward. As shown in Tab. 4, all these approaches yield noticeable improvements over the base model on audio–visual understanding benchmarks, *confirming the effectiveness of $\mathcal{D}$ after our data preparation pipeline.*

However, the performance gains of these baselines are consistently smaller than those achieved by OmniVideo-R1. On Daily-Omni, our method surpasses the second-best Vanilla RL by **12.0%** (82.8 vs. 73.9), and on WorldSense it outperforms the second-best CoT SFT by **11.1%** (65.8 vs. 59.2). *These ablation results further validate the effectiveness and superiority of our training paradigm.*

### 4.4. Generalization to Other Backbones.

To verify that OmniVideo-R1 is not limited to a specific architecture, we apply our framework to two additional omni backbones, using GSPO or GRPO depending on whether the architecture is MoE-based or dense. As shown in Tab. 5, OmniVideo-R1 consistently improves performance across all three backbones. Specifically, OmniVideo-R1 yields a **32.1%** relative improvement on Qwen2.5-Omni-7B (Xu et al., 2025a) (38.7 vs. 29.3) and a **21.9%** gain on MiniCPM-

| Method | Video-MME | MLVU(Dev) | LVBench |
|---|---|---|---|
| GPT-4o | 71.9 | 64.6 | 30.8 |
| Gemini-2.0-Flash | 72.4 | 71.0 | 57.9 |
| Gemini-2.5-Pro (Google, 2024) | 86.9 | 81.2 | 69.2 |
| VideoLLaMA3-7B (Zhang et al., 2025) | 66.2 | 73.0 | 45.3 |
| InternVideo2.5-8B (Wang et al., 2025c) | 65.1 | 72.8 | 46.4 |
| Qwen2.5-VL-7B (Bai et al., 2025b) | 65.1 | 70.2 | 45.3 |
| Qwen2.5-VL-72B (Bai et al., 2025b) | 73.3 | 74.6 | 47.3 |
| video-SALMONN 2+-7B (Tang et al., 2025) | 73.4 | 73.6 | 49.7 |
| Qwen3-Omni-30B-A3B-Instruct (Xu et al., 2025c) | 70.5 | 75.2 | 50.2 |
| Qwen3-Omni-30B-A3B-Thinking (Xu et al., 2025c) | 69.7 | 72.9 | 49.0 |
| OmniVideo-R1 | **73.6** | 74.1 | **51.9** |

*Table 3.* Performance of different methods on various visual-only benchmarks, including Video-MME (Fu et al., 2025a), MLVU (Zhou et al., 2025a) and LVBench (Wang et al., 2025b). The **best** is highlighted and the second-best is underlined.

| Method | OmniVideoBench | Daily-Omni | WorldSense |
|---|---|---|---|
| QA SFT | 39.1 | 69.9 | 57.4 |
| CoT SFT | 42.2 | 73.1 | 59.2 |
| Vanilla RL | 41.5 | 73.9 | 58.0 |
| Full | **44.8** | **82.8** | **65.8** |

*Table 4.* Performance on different training strategies in terms of Qwen3-Omni-30B-A3B-Instruct. The **best** is highlighted.

o-2.6 (Yao et al., 2024) (36.2 vs. 29.7).

| Backbone | Method | OmniVideoBench |
|---|---|---|
| Qwen2.5-Omni-7B | Baseline | 29.3 |
| Qwen2.5-Omni-7B | OmniVideo-R1 | **38.7** |
| MiniCPM-o-2.6 | Baseline | 29.7 |
| MiniCPM-o-2.6 | OmniVideo-R1 | **36.2** |

*Table 5.* Accuracy across different omni backbones.

## 4.5. Case Analysis

We further present qualitative case for QI-only training, and our QI+MA (OmniVideo-R1) training. As shown in Fig. 5, QI training yields strong reasoning behavior; however, in some cases the model overlooks critical audio cues, resulting in inaccurate inferences. In contrast, our QI+MA, first establishing the desired reasoning behavior and then booming deeper audio-visual reasoning, enables the model to better exploit both audio and visual evidence. More cases are illustrated in the Appendix.

## 4.6. Ablation Study

**Component Removal.** We first perform ablations on various designs (w/o $r_{attn}$, $r_{intent}$, or QI stage) in Tab. 6. The results suggest that the observed performance gains are mainly attributable to two factors: *(1) $r_{intent}$, which encourages accurate grounding of the primary cues, and (2) modality-attentive training, which further strengthens the model's ability to perform comprehensive audio-visual reasoning.*

It can be observed that *performing MA stage training alone can bring substantial improvements* (as shown in "w/o QI" setting in the table). For instance, on OmniVideoBench,

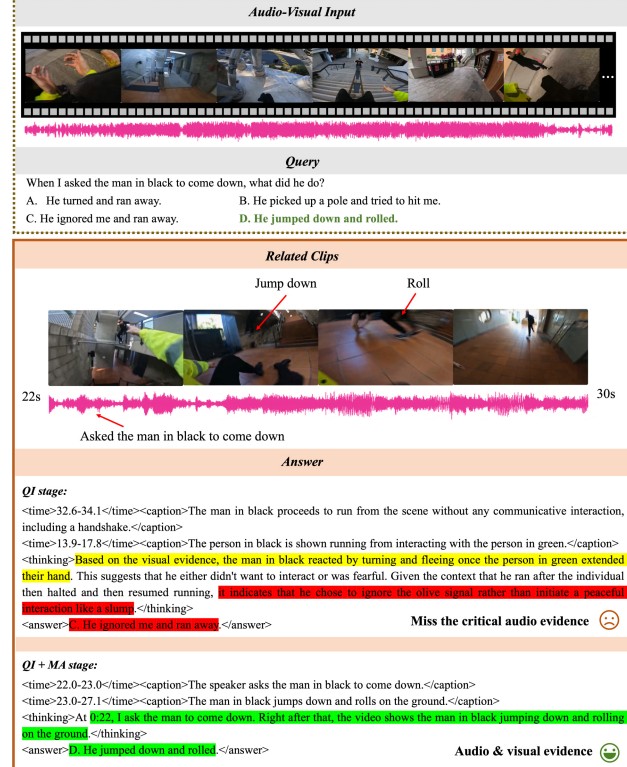

*Figure 5.* Visualization of the results obtained from the training of QI, and QI+MA. Red highlights the incorrect text, while green highlights the correct text. Yellow highlights the model overemphasizes one modality while neglecting cues from the other modality.

| Method | OmniVideoBench | Daily-Omni | WorldSense |
|---|---|---|---|
| w/o $r_{intent}$ | 38.4 | 75.9 | 55.1 |
| w/o $r_{attn}$ | 43.7 | 82.1 | 65.5 |
| w/o QI (10K) | 41.6 | 76.1 | 58.6 |
| w/o QI (88K) | 41.0 | 76.9 | 59.2 |
| w/o MA | 43.6 | 82.0 | 65.3 |
| w/o audio input | 37.6 | 68.7 | 50.3 |
| w. timestamps | 43.4 | 81.7 | **66.1** |
| Base model | 37.0 | 63.6 | 54.0 |
| Full | **44.8** | **82.8** | 65.8 |

*Table 6.* Performance on different ablated settings in terms of Qwen3-Omni-30B-A3B-Instruct. The **best** is highlighted.

this strategy yields a **12.4%** gain over the base model (41.6 vs. 37.0). Furthermore, QI stage training ("w/o MA" setting in the table) also significantly improved the model's capability, yielding a **17.8%** gain over the base model (43.6 vs. 37.0). Removing $r_{intent}$ or $r_{attn}$ both results in certain performance drop.

**Input Configuration.** We further ablate the impact of input configuration in Tab. 6. Specifically, we use OmniVideo-R1 trained with both audio and video, but perform inference under a "w/o audio input" setting. We observe a performance drop on WorldSense (50.3 vs. 54.0), but a slight improvement on Daily-Omni (68.7 vs. 63.6). This mixed behavior can be attributed to two factors: (1) when the evaluation benchmark inherently relies on audio, the *mismatch* between training (w. audio) and inference (w/o audio) naturally leads to degraded performance; (2) owing to the enhanced reasoning capability and *robust grounding of key visual cues*, the model can actually *perform better on tasks where audio is non-essential or visual information alone is sufficient.*

Moreover, many recent methods introduce explicit temporal cues by overlaying timestamps (Ge et al., 2025). While this can strengthen temporal perception, it simultaneously *occludes part of the original visual content*. In contrast, our $r_{intent}$ reward inherently promotes temporal correction during training (e.g., *inaccurate temporal grounding directly degrades caption quality*), endowing OmniVideo-R1 with an implicit sense of time. As a result, our method is insensitive to such numeric overlays ("w. timestamps"), exhibiting only marginal performance differences.

**MA Hyperparameter $\alpha$.** Tab. 7 ablates the attention reward magnitude $\alpha$. A moderate value ($\alpha = 0.3$) yields the best results. That is, an excessively small value provides insufficient incentive for cross-modal exploitation, whereas an overly large one can overshadow other reward signals.

| $\alpha$ | OmniVideoBench | Daily-Omni |
|------|------|------|
| 0.1 | 44.3 | 82.3 |
| 0.3 | **44.8** | **82.8** |
| 0.8 | 44.6 | 82.5 |
| 1.0 | 44.6 | 82.4 |

*Table 7.* Effect of the attention reward hyperparameter $\alpha$.

**MA-Stage Data Scaling.** Tab. 8 examines the effect of varying the amount of MA-stage data within OmniVideo-R1 pipeline. The performance continues to improve as the MA data scale increases (e.g., 1K MA samples improves performance from 43.6 to **44.0**), although the marginal gain gradually diminishes.

**Effect of Sampled Frames.** Tab. 9 examines the impact of FPS_MAX_FRAMES on OmniVideoBench, with the same value used in both training and inference. On the one hand, events in most videos are not sufficiently dense;

| Stage | MA Data | OmniVideoBench | Daily-Omni |
|------|------|------|------|
| w/o MA | - | 43.6 | 82.0 |
| QI+MA | 1K | 44.0 | 82.4 |
| QI+MA | 5K | 44.5 | 82.6 |
| QI+MA | 12K | 44.8 | 82.8 |
| QI+MA | 30K | 44.9 | 82.9 |
| QI+MA | 50K | 45.0 | 83.0 |
| QI+MA | 88K | 45.0 | 83.1 |

*Table 8.* Effect of MA-stage data scale in OmniVideo-R1 pipeline.

typically, only one event or shot appears within a short period, so a lower sampling rate may not necessarily miss critical information. On the other hand, as shown above, after balancing accuracy and computational cost, we chose **64** in the current paper.

| Duration | 32 | 64 | 128 | 256 |
|------|------|------|------|------|
| (0,1] min | 49.1 | 53.8 | 54.4 | 49.1 |
| (1,5] min | 40.9 | 43.5 | 41.8 | 42.6 |
| (5,10] min | 42.5 | 43.9 | 43.9 | 46.1 |
| (10,30] min | 40.2 | 41.4 | 42.6 | 44.5 |
| Avg. | 42.5 | 44.8 | 44.6 | 45.0 |

*Table 9.* Performance comparison on OmniVideoBench across varying FPS_MAX_FRAMES configurations.

## 5. Conclusion and Limitation

**Conclusion.** In this paper, we propose OmniVideo-R1, a query-intensive deep fusion framework for audio-visual reasoning. Our training pipeline consists of two stages. First, without relying on any process-level annotations, we encourage the model to "think with omnimodal cues" by learning in a self-supervised manner grounded in intermediate time-caption pairs. Second, we explicitly enhance cross-modal fusion by contrasting the model's learning under full audio-visual input versus single-modality input, thereby improving its ability to build coherent multimodal representations. Extensive experiments show that OmniVideo-R1 consistently outperforms prior methods on multiple benchmarks, laying a solid foundation for future work in audio-visual reasoning.

**Limitation & Future Work.** (1) Current methods still rely on outcome-based ground-truth for training. Exploring how to effectively strengthen the model in the absence of ground-truth could be an important direction for future research. (2) The multimodal training paradigm is not restricted to audio–visual inputs. With our proposed query intention and modality attention, it can extend to more modalities (e.g., 3D).

## Acknowledgments

This work was supported by the National Natural Science Foundation of China under contract No. 62171256.

## Impact Statement

This paper presents work whose goal is to advance the field of machine learning. There are many potential societal consequences of our work, none of which we feel must be specifically highlighted here.

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

In this Appendix, we provide more technical details, including 1) detailed descriptions of training dataset and benchmarks in Sec. A; 2) the specific prompts used in our experiments in Sec. B; 3) additional case study in Sec. C; 4) additional experiments and analysis in Sec. D; 5) time cost in Sec. E; and 6) the hyperparameter configurations of training settings in Sec. F.

## A. Dataset

### A.1. Training Dataset

As illustrated in Fig. 4, we primarily perform a three-stage data filtering and selection pipeline to obtain high-quality audio–video data. To more clearly present the distribution of the processed data, we report descriptive statistics of the resulting training dataset as shown in Fig. 6. That is, our dataset comprises 16 categories with varying numbers of samples, ranging from 35 to 34598. The questions with audio–video are of high quality and exhibit substantial diversity in content.

### A.2. Benchmarks

#### A.2.1. AUDIO-VISUAL BENCHMARKS

**OmniVideoBench (Li et al., 2025a):** a large-scale, carefully curated benchmark for evaluating synergistic audio–visual reasoning, with particular emphasis on *modality complementarity and logical coherence*. It contains 1000 high-quality question–answer pairs, derived from 628 diverse videos spanning from a few seconds to 30 minutes.

**Daily-Omni (Zhou et al., 2025b):** an audio–visual question answering dataset containing 684 *daily-life videos* from diverse sources, rich in both auditory and visual cues, and providing 1197 multiple-choice QA pairs spanning 6 major tasks.

**WorldSense (Benchekroun et al., 2023):** a benchmark emphasizing *omnimodal collaboration*, with strongly coupled audio–video tasks that require synergistic multimodal perception. It contains 1662 synchronized audio–visual videos across 8 domains and 67 subcategories, and 3172 multiple-choice QA pairs covering 26 tasks for comprehensive evaluation

**IntentBench (Yang et al., 2025b):** a benchmark designed to evaluate models' understanding of complex *human intentions and emotions*, comprising 633 videos and 2689 questions grounded in both auditory and visual cues.

**VideoHolmes (Cheng et al., 2025):** a Sherlock Holmes–inspired benchmark for evaluating *complex video reasoning* in MLLMs, featuring 1837 questions from 270 annotated suspense short films across seven tasks, each requiring models to connect dispersed visual clues and underlying causal events.

#### A.2.2. VISUAL-ONLY BENCHMARKS

**Video-MME (Fu et al., 2025a):** the first *full-spectrum, multimodal evaluation benchmark* for MLLMs in video analysis, covering 6 major visual domains and 30 subdomains, with 900 videos ranging from 11 seconds to 1 hour (totaling 254 hours) and 2700 QA pairs.

**MLVU (Zhou et al., 2025a):** the benchmark focuses on *long videos and diversity in both video types and evaluation tasks*, with durations ranging from 3 minutes to 2 hours and a total of 9 different evaluation tasks. In this paper, *we use its dev subset for evaluation.*

**LVBench (Wang et al., 2025b):** a benchmark specifically designed for *ultra-long video understanding spanning several hours*, aimed at testing MLLMs' long-term memory and extended comprehension abilities. It contains 103 videos and 1549 question–answer pairs in total.

## B. Instruction Details

### B.1. OmniVideo-R1

As illustrated in Fig. 8, we employ a specialized system prompt as well as the fixed suffix to the user prompt for OmniVideo-R1. In this way, the model can undergo training for specific reasoning behaviors starting from zero-RL.

### B.2. Data Preparation

In terms of data preparation, in the last stage, we perform categorization based on Qwen-3-32B (Yang et al., 2025a) as the instruction shown in Fig. 7, dividing the data into 16 categories. The final results of this categorical analysis are shown in Fig. 6. On the other hand, we use the prompt in Fig. 9 for data quality assessment.

### B.3. Consistency Judger

As shown in Fig. 2, the consistency judger is mainly used for rewarding the consistency of time-caption pairs. It is primarily based on Qwen3-VL-235B-A22B-Instruct (Bai et al., 2025a) for scoring, and the corresponding prompt is illustrated in Fig. 10.

### B.4. Completeness Evaluator

Meanwhile, as shown in Fig. 2, the completeness evaluator is mainly used to assess the completeness of multiple audio–video segments produced by query-intensive grounding. It is also based on Qwen3-VL-235B-A22B-Instruct (Bai et al., 2025a) for scoring, and the corresponding prompt is illustrated in Fig. 11.

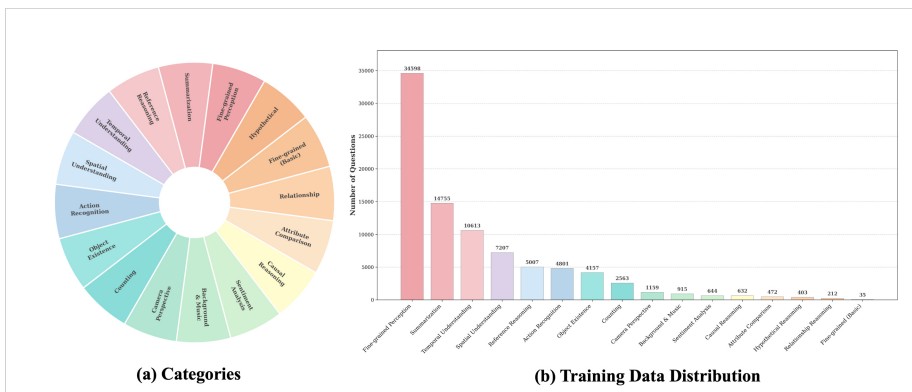

*Figure 6.* (a) Our training data covers 16 categories. (b) Number of questions in terms of each category.

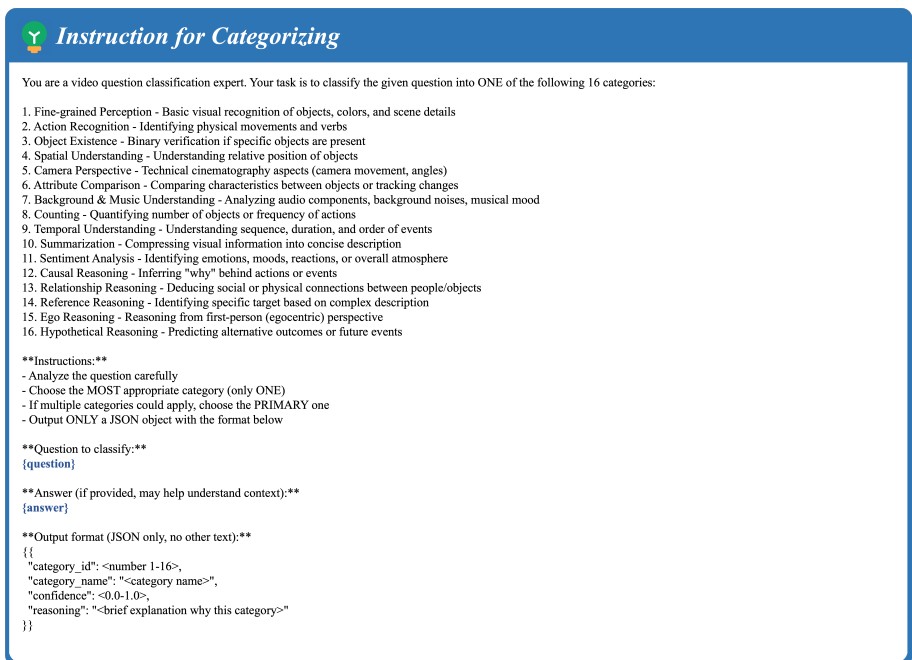

*Figure 7.* Instruction for data categorizing in data preparation.

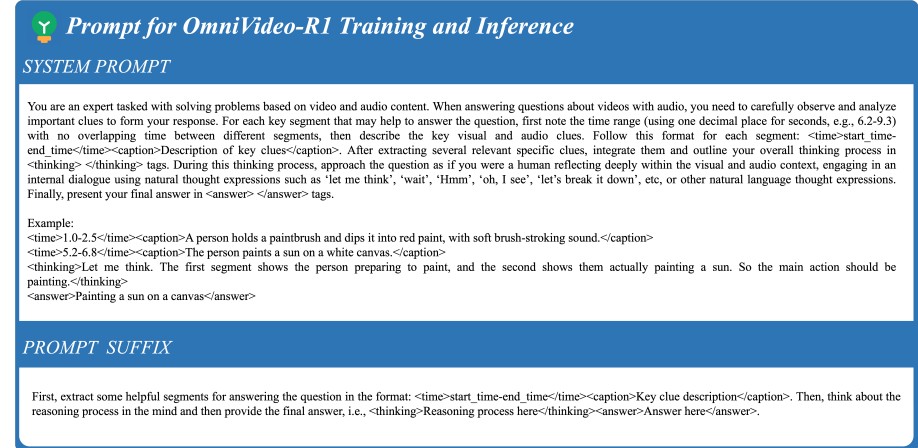

*Figure 8.* System prompt and user prompt suffix for OMNIVIDEO-R1 reasoning.

## 🌱 *Instruction for Data Quality Assessment*

**Role Definition:**
You are a senior VideoQA dataset quality auditor. Your task is to rigorously review the given 【Question】 and 【Annotated Answer】, combined with video/audio context (if you have visual capability, otherwise based on logical reasoning), to identify low-quality, logically flawed, or "dirty data" that can be answered without audio-visual information.

**Input Information:**
- **【Question】**: {problem}
- **【Annotated Answer】**: {solution}

---

**Review Dimensions and Scoring Criteria (10 points max per dimension):**

#### Dimension 1: Video Dependency
**Core Detection Point:** Is watching the video necessary to answer the question?
- **Scoring Criteria:**
  - **0-2 points (Must Filter):** The answer can be derived solely through common sense, grammatical clues in the question, or the exclusivity of options without watching the video (e.g., the answer is contained in the question, or there are obvious common-sense errors in the options).
  - **3-6 points (Questionable):** The question is vague, and one can likely guess correctly through inference; video information only provides weak assistance.
  - **7-10 points (High Quality):** Must combine specific visual details from the video (actions, objects, scenes, text, colors, movements) to answer; cannot be guessed without watching.

#### Dimension 2: Audio Dependency
**Core Detection Point:** Is listening to the audio necessary to answer the question?
- **Scoring Criteria:**
  - **0-2 points (Not Required):** The question does not require any audio information; it can be answered purely from visual content or common sense.
  - **3-6 points (Helpful but Optional):** Audio provides supplementary context (e.g., background music mood, ambient sounds) but is not essential for answering.
  - **7-10 points (Essential):** Must listen to the audio to answer correctly (e.g., questions about speech content, sound effects, music, dialogue, narration).

#### Dimension 3: Question Logic & Rationality
**Core Detection Point:** Is the question valid? Is the logic self-consistent?
- **Scoring Criteria:**
  - **0-2 points (Must Filter):** The question not only has grammatical errors but also contains logical paradoxes; or the question asks about content that typically cannot be obtained from video (e.g., asking about the filmmaker's psychological state).
  - **3-6 points (Average):** The question is wordy, with unclear references (e.g., "that thing"), but barely understandable.
  - **7-10 points (High Quality):** The question is clear, precise, with unambiguous references, conforming to natural human questioning habits.

#### Dimension 4: Answer Accuracy & Spatial/Temporal Precision
**Core Detection Point:** **Is the answer correct? **Especially, are directions (left/right), colors, quantities, and temporal order (before/after) accurate?**
- **Scoring Criteria:**
  - **0-2 points (Must Filter):** The answer is clearly wrong; **left/right directions are reversed**; color/quantity descriptions severely contradict common sense or visual features; the answer is irrelevant to the question.
  - **3-6 points (Questionable):** The answer is partially correct but contains redundant information or slight ambiguity in direction/detail descriptions (e.g., "left" described as "beside").
  - **7-10 points (High Quality):** The answer is precise, accurately capturing key visual/audio information from the video, with completely correct direction and detail descriptions.

---

**Composite Score Calculation Formula:**
`Composite Score = (Video Dependency * 0.3) + (Audio Dependency * 0.2) + (Question Logic * 0.25) + (Answer Accuracy * 0.25)`

**Filtering Threshold:**
- If **Video Dependency score < 4 points**, then `should_filter` is `true` (question doesn't need video).
- If **Question Logic score < 4 points**, then `should_filter` is `true`.
- If **Answer Accuracy score < 4 points**, then `should_filter` is `true`.
- If **composite score < 5 points**, then `should_filter` is `true`.
- Note: Low Audio Dependency alone does NOT trigger filtering (many valid questions don't require audio).

---

**Output Format (JSON):**
Please strictly output in the following JSON format, without any text other than the JSON:

```json
{{
  "scores": {{
    "video_dependency_score": 0,
    "audio_dependency_score": 0,
    "question_logic_score": 0,
    "answer_accuracy_score": 0,
    "composite_score": 0
  }},
  "dimensions_analysis": {{
    "video_dependency_reason": "Explain whether watching the video is necessary and why (e.g., requires seeing specific actions, objects, colors, movements).",
    "audio_dependency_reason": "Explain whether listening to audio is necessary and why (e.g., requires hearing speech, music, sound effects, or audio is not relevant).",
    "question_logic_reason": "Evaluate the rationality and clarity of the question.",
    "answer_accuracy_reason": "Evaluate the correctness of the answer, especially noting any left/right direction, color, or counting errors."
  }},
  "should_filter": true,
  "filter_primary_reason": "If filtered, specify the primary reason (e.g., no video needed/direction error/logical confusion); if not filtered, fill null."
}}
```

*Figure 9.* Instruction for data quality assessment in data preparation.

## 💡 *Prompt for Consistency Judger*

**SYSTEM PROMPT**

You are an expert evaluator for video content description. Your primary task is to verify the factual correctness of the statements made in the caption against the video content.

Evaluation Criteria:
1. **Fact Verification:** Every visual claim, action, and object mentioned in the caption is explicitly supported by the video footage.
2. **Descriptive Accuracy:** The specific attributes described in the text (such as colors, directions, counts, and sequences) match the actual visual elements.
3. **Absence of Fabrications:** The caption does not contain hallucinations or descriptions of objects and events that are absent from the video.
4. **Logical Audio Consistency:** Audio descriptions appearing in the text are considered accurate as long as they are contextually plausible given the visual scene (e.g., "people talking" is valid if mouths move). They are only penalized if they are blatantly contradictory to the visual context.

Scoring Guidelines:
- 1.0: All statements in the caption are completely accurate and visually verified.
- 0.7-0.9: The caption is mostly accurate, with only very minor phrasing issues or negligible details that deviate slightly from the video.
- 0.4-0.6: The caption describes the correct general topic, but contains specific claims that are false or not present in the video (partial hallucination).
- 0.1-0.3: The caption contains major fabrications, describing actions or objects that clearly do not exist in the video.
- 0.0: The caption describes a completely different scenario or is irrelevant.

You must respond with ONLY a JSON object, no other text or explanation:
{"score": <float between 0.0 and 1.0>, "reason": "<brief explanation for the score>"}"

**USER PROMPT**

Video segment is provided above.

Caption to evaluate: {caption}

Evaluate whether the descriptions in this caption are accurate based on the video content. Output ONLY the JSON object, no other text:
{{"score": <float>, "reason": "<explanation>"}}

*Figure 10.* System prompt and user prompt for consistency judger.

## 💡 *Prompt for Completeness Evaluator*

**SYSTEM PROMPT**

You are an expert evaluator for video-based question answering. Your task is to evaluate the quality of selected video segments based on their ability to support a specific question-answer pair. The input video consists of stitched clips.

Please evaluate the segments according to the following standards:
1. Content Sufficiency & Inference: The segments provide comprehensive visual evidence to support the answer. For questions requiring audio information, the visual context (such as a speaking person, subtitles, specific gestures, or sound-making actions) logically implies and supports the content of the answer.
2. Temporal Precision: The segments are tightly trimmed to include only the key moments, avoiding excessive padding, long periods of inactivity, or redundant scenes.
3. Completeness: The segments capture all necessary steps or details required to derive the solution, ensuring the visual flow is consistent with the answer.

Scoring Guidelines:
- 1.0: The segments contain all necessary visual evidence (or logical visual context for audio cues) and are precisely trimmed with minimal redundancy.
- 0.7-0.9: The segments provide sufficient visual context but include minor irrelevant footage or slightly loose trimming.
- 0.4-0.6: The segments contain the core visual context but suffer from significant redundancy or lack clear visual logical links to the answer.
- 0.1-0.3: The segments provide only partial visual info or the visual context poorly matches the audio-based answer.
- 0.0: The segments are completely irrelevant or fail to visually show the source of the answer.

You must respond with ONLY a JSON object, no other text or explanation:
{"score": <float between 0.0 and 1.0>, "reason": "<brief explanation for the score>"}

**USER PROMPT**

The video segments shown above are the key segments selected by the model.

Question: {question}

Ground Truth Answer: {solution}

Evaluate whether these segments provide sufficient visual evidence (or visual logic for audio questions) to support the answer while remaining concise. Output ONLY the JSON object, no other text:
{{"score": <float>, "reason": "<explanation>"}}

*Figure 11.* System prompt and user prompt for completeness evaluator.

## C. Additional Case Study

We provide an additional case as illustrated in Fig. 12. The results show that QI training tends to introduce redundant grounding, as its primary objective is to shape reasoning behavior. Our QI+MA further use MA to maximize the utilization of audio-visual cues.

## D. Additional Experiments and Analysis

### D.1. Reliability of External Judges

To assess the reliability of external models used in data filtering and reward computation, we conduct systematic human verification studies.

**Data Construction.** We randomly sampled **100** examples and manually verified the judgments produced by Gemini-2.5-Pro using a binary pass/fail criterion. As shown in Tab. 10, all four scoring dimensions achieve at least **97%** accuracy. We also evaluated the 15-category classification by Qwen-3-32B, which achieved an accuracy of **0.96**.

| Dimension | Accuracy |
|---|---|
| Video dependency | 1.00 |
| Audio dependency | 0.98 |
| Question logic | 0.98 |
| Response accuracy | 0.97 |

*Table 10.* Human verification accuracy of Gemini-2.5-Pro judgments on data construction dimensions (100 sampled examples).

**Reward Computation.** Using the 97 response-correct samples from the above subset (covering 485 time-caption pairs), we evaluated the reward scores produced by Qwen3-VL-235B-A22B-Instruct via the same binary pass/fail protocol. As shown in Tab. 11, all reward components achieve accuracies of at least **94%**.

| Reward | Accuracy |
|---|---|
| Consistency | 0.961 |
| Completeness | 0.940 |
| Answer | 0.990 |

*Table 11.* Human verification accuracy of reward scores produced by Qwen3-VL-235B-A22B-Instruct.

**Cross-Judge Agreement.** We further compared the agreement between Qwen3-VL-235B-A22B-Instruct and alternative judge models using the same protocol. As shown in Tab. 12, high Cohen's $\kappa$ (Cohen's Kappa) values indicate strong inter-judge agreement, suggesting that *our OmniVideo-R1 framework does not rely on an exceptionally strong omnimodal judger.*

### D.2. Error Analysis

We analyzed all failure cases of OmniVideo-R1 on OmniVideoBench and randomly sampled **50** incorrect exam-

| Judge Pair | Cons. $\kappa$ | Comp. $\kappa$ | Ans. $\kappa$ |
|---|---|---|---|
| Gemini-2.5-Pro vs. Qwen3-VL | 0.78 | 0.82 | 0.97 |
| GPT-4.1 vs. Qwen3-VL | 0.84 | 0.86 | 0.98 |

*Table 12.* Pairwise Cohen's $\kappa$ agreement between Qwen3-VL-235B-A22B-Instruct and alternative judge models. Cons. denotes consistency, comp. denotes completeness, and ans denotes answer.

ples for manual inspection, with each example potentially belonging to multiple error categories. As shown in Tab. 13, temporal grounding errors dominate (**72%** of inspected failures), primarily manifesting as: (i) failure to identify the critical time span, and (ii) cases where the generated caption contains correct information but is associated with an incorrect temporal segment. This suggests that improving temporal grounding accuracy is the most promising direction for future gains.

| Error Type | Count |
|---|---|
| Temporal grounding error | 36 |
| Reasoning error | 14 |
| Caption error | 8 |
| Format error | 0 |

*Table 13.* Distribution of error types on 50 sampled failure cases from OmniVideoBench.

### D.3. Extended Ablation Studies

**Reward Weight Sensitivity.** As shown in Tab. 14, we report the effect of varying reward weights in both QI and MA stages. Reducing $r_{\mathrm{intent}}$ has the largest impact, confirming that query-intensive grounding is the primary driver of performance gains.

| Stage | $r_{\mathrm{fmt}}$ | $r_{\mathrm{ans}}$ | $r_{\mathrm{int}}$ | $r_{\mathrm{attn}}$ | OmniVideoBench | Daily-Omni |
|---|---|---|---|---|---|---|
| QI | 1 | 1 | 1 | - | **43.6** | **82.0** |
| QI | 0.5 | 1 | 1 | - | 43.5 | 81.7 |
| QI | 1 | 0.5 | 1 | - | 42.7 | 81.4 |
| QI | 1 | 1 | 0.5 | - | 41.0 | 80.3 |
| QI | 1 | 1 | 0 | - | 38.4 | 75.9 |
| QI+MA | 1 | 1 | - | 1 | **44.8** | **82.8** |
| QI+MA | 0.5 | 1 | - | 1 | 44.6 | 82.5 |
| QI+MA | 1 | 0.5 | - | 1 | 44.0 | 82.3 |
| QI+MA | 1 | 1 | - | 0.5 | 43.9 | 82.3 |
| QI+MA | 1 | 1 | - | 0 | 43.7 | 82.1 |

*Table 14.* Reward weight sensitivity analysis. $r_{\mathrm{fmt}}$, $r_{\mathrm{ans}}$, $r_{\mathrm{int}}$, and $r_{\mathrm{attn}}$ denote format, answer, intent, and attention rewards.

## E. Time Cost

Tab. 15 reports the average wall-clock time per rollout for each reward component under our 8×H20 training setup (judge deployed on 64×H20). The main bottlenecks are judge-model inference and video clipping/concatenation. For $r_{\mathrm{cons}}$, parallel evaluation over multiple time-caption pairs reduces runtime from ∼19.65s to ∼6.21s.

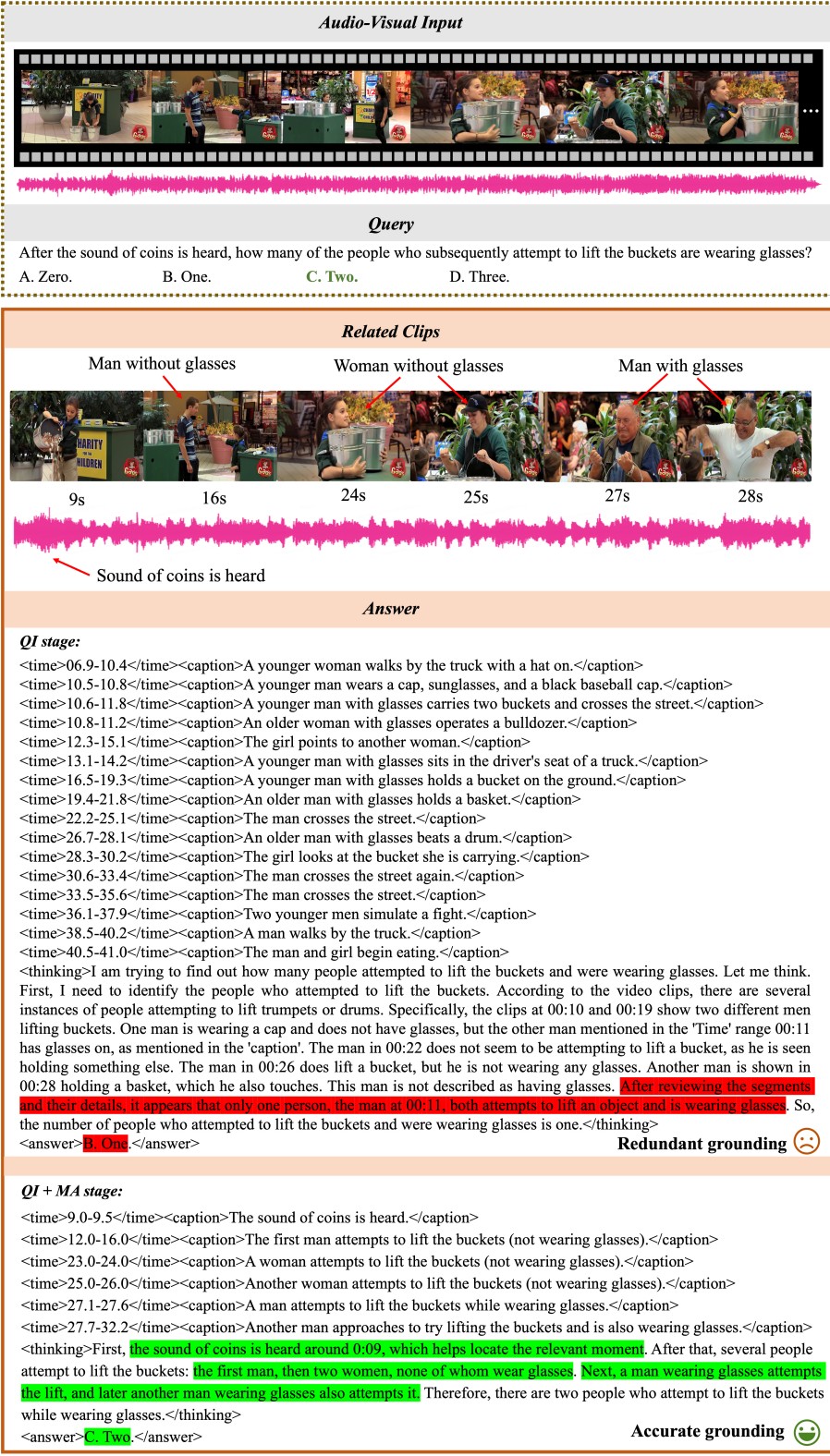

*Figure 12.* Visualization of the results obtained from the training of QI, and QI+MA. Red highlights the incorrect text, while green highlights the correct text.

| Reward | Granularity | Time / Rollout |
|---|---|---|
| $r_{\text{format}}$ | Sequence-level | $\sim$1.62s |
| $r_{\text{cons}}$ | Timestamp-level | $\sim$6.21s |
| $r_{\text{comp}}$ | Sequence-level | $\sim$7.97s |
| $r_{\text{ans}}$ | Sequence-level | $\sim$4.70s |
| $r_{\text{attn}}$ | Sequence-level | $\sim$8.06s |

*Table 15.* Average wall-clock time per rollout for each reward.

## F. Implementation Details

We set more of the key hyperparameters as follows: FPS_MAX_FRAMES 64 to cap the number of frames per sample, lr_warmup_fraction 0.05 to gradually ramp up the learning rate at the start of training, $\epsilon$ $3 \times 10^{-4}$ and $\epsilon_{\text{high}}$ $4 \times 10^{-4}$ as clipping thresholds, KL regularization coefficient $\beta$ 0.03 to penalize large deviations from the reference policy, and moe_aux_loss_coeff $10^{-3}$ to weight the auxiliary load-balancing loss for the mixture-of-experts.

