# OpenReview forum: "OmniVideo-R1: Reinforcing Audio-visual Reasoning with Query Intention and Modality Attention"
_ICML.cc/2026/Conference — ICML 2026 regular_

### Official Review · Reviewer_VYQA · 2026-02-18

**Soundness:** 2
**Presentation:** 3
**Significance:** 3
**Originality:** 3
**Overall Recommendation:** 5
**Confidence:** 4

**Summary:**

This paper proposes OmniVideo-R1. It points out that, due to imbalanced training data distributions, omni models suffer from degraded visual capability compared with their homogeneous VL counterparts. To address this issue, this paper introduces a two-stage reinforcement learning post-training strategy to correct the bias. In the first stage, Query Intention (QI) is employed to teach the model to localize key audiovisual segments. In the second stage, Modality Attention (MA) fusion is introduced; inspired by contrastive learning, it encourages the model to achieve higher confidence under mixed-modality inputs. The reported performance is highly impressive, substantially outperforming existing models.

**Compliance With Llm Reviewing Policy:**

Affirmed.

**Final Justification:**

The author's rebuttal make the score changing from 4 to 5. The reviewer would like to accept this paper.

**Key Questions For Authors:**

1. Does the authors plan to release the model weights and training data? Open-sourcing these resources would substantially enhance the paper’s impact and benefit the broader research community.

2. In Section 4.3, this paper reports an ablation study using the GRPO algorithm under a think-then-answer paradigm, with a mixture of format and soft-response scores as the reward. However, to the best of my understanding, GRPO is not inherently restricted to this specific formulation. Since the proposed QI and MA strategies are trained with the GSPO algorithm, it seems plausible that GSPO could be replaced by GRPO in this framework. Could the authors conduct such an experiment and provide a detailed comparison between GRPO and GSPO in practical settings, clarifying their respective advantages and limitations? If this understanding is inaccurate, clarification would be appreciated.

3. Is 'FPS_MAX_FRAMES = 64' sufficient for handling long videos in many of the evaluated benchmarks? For example, for a 10-minute video, this corresponds to approximately 0.1 FPS. Several benchmarks used in this paper include a substantial number of videos longer than 10 minutes, and even exceeding 30 minutes. It would be helpful to discuss whether this temporal sampling strategy adequately preserves critical information in long-form video scenarios.

**Limitations:**

yes

**Strengths And Weaknesses:**

Strengths:
1. The two-stage reinforcement learning strategy is thoughtfully designed. The model first learns to localize key audiovisual segments through QI, and is then encouraged to achieve higher confidence under mixed-modality inputs via a contrastive learning-inspired MA objective. This progressive training paradigm is well-motivated and technically sound.

2. The model demonstrates strong performance across multiple audiovisual benchmarks, substantially outperforming existing approaches.

Weaknesses:
1. In the Introduction, this paper highlights that Qwen3-series omni models exhibit degraded visual performance compared to their VL counterparts on vision-centric benchmarks (e.g., MMStar and MathVista_mini), and claims that OmniVideo-R1 “unlocks true omnimodal synergy.” However, Table 3 does not provide a direct comparison between OmniVideo-R1 and Qwen3-VL on visual benchmarks. Additional experimental results would better substantiate this claim and strengthen the empirical support for the paper’s central argument.

2. It remains unclear how the model behaves when errors occur during the reasoning process, e.g., if it grounds incorrect temporal segments or generates hallucinated captions. More quantitative or qualitative analyses of such failure cases would provide deeper insights into the proposed method and offer valuable guidance for future work.

---

> ### Author Rebuttal · Authors · 2026-03-30
>
> We sincerely thank the reviewer for the thoughtful and constructive feedback. We have carefully considered all comments and will revise the paper accordingly.
>
> ### **Q1. Visual comparison with Qwen3-VL**
>
> Thank you for this important suggestion. We have added a comparison against Qwen3-VL-30B-A3B on visual benchmarks:
>
> | Method | MMStar ↑ | MathVista_mini ↑ |
> | --- | :---: | :---: |
> | Qwen3-VL-30B-A3B | 72.1 | 80.1 |
> | Qwen3-Omni-30B-A3B | 68.5 | 75.9 |
> | **OmniVideo-R1** | **73.1** | **80.8** |
>
> These results provide more direct evidence for our motivation: the base omni model is weaker than the corresponding VL model on visual benchmarks, while OmniVideo-R1 helps narrow this gap and, in this setting, even surpasses it.
>
> ### **Q2. Failure cases and error analysis**
>
> We analyzed all failure cases on OmniVideoBench and randomly sampled 50 examples for manual inspection. We first categorized them with Gemini-3-Pro and then manually corrected the labels. Each example may belong to multiple categories.
>
> | Error Type | Number ↓ |
> | --- | :---: |
> | Temporal grounding error | 36 |
> | Caption error | 8 |
> | Reasoning error | 14 |
> | Format error | 0 |
>
> These results suggest that errors in locating the key temporal segment, or localizing redundant segments, account for the majority of failures (78%). The main patterns are: (i) failure to identify the critical time span, and (ii) cases where the caption contains the correct key information but the localized time segment is incorrect. In the revised paper, we will provide a more detailed discussion and highlight this as an important direction for future work.
>
> ### **Q3. Open-source plan**
>
> Yes. We plan to release the training and evaluation code, checkpoints, as well as dataset. We will also release the relevant training and data construction details to improve reproducibility for community.
>
> ### **Q4. Comparison between GSPO and GRPO**
>
> We further trained our method with GRPO, on OmniVideoBench, OmniVideo-R1 GSPO vs. OmniVideo-R1 GRPO = **44.8** VS. 42.8. On Daily-Omni, OmniVideo-R1 GSPO vs. OmniVideo-R1 GRPO = **82.8** VS. 77.6. In our experiments, under MoE settings, GSPO was more stable than token-level GRPO and achieved better performance.
>
> For a fair comparison, we also included a vanilla RL baseline using GSPO. Compared with OmniVideo-R1, it achieves `41.9` vs. `44.8` on OmniVideoBench and `74.6` vs. `82.8` on Daily-Omni. We hope these results help clarify the effectiveness of our method. We will revise the wording in the paper and add these results as explicit evidence.
>
> ### **Q5. Choice of `FPS_MAX_FRAMES`**
>
> We appreciate this concern. To further examine the effect of `FPS_MAX_FRAMES`, we report accuracy on OmniVideoBench under different settings, with the same value used in both training and inference:
>
> | Frames | 32 | 64 | 128 | 256 |
> | --- | :---: | :---: | :---: | :---: |
> | `(0,1]` min Acc. ↑ | 49.1 | 53.8 | 54.4 | 49.1 |
> | `(1,5]` min Acc. ↑ | 40.9 | 43.5 | 41.8 | 42.6 |
> | `(5,10]` min Acc. ↑ | 42.5 | 43.9 | 43.9 | 46.1 |
> | `(10,30]` min Acc. ↑ | 40.2 | 41.4 | 42.6 | 44.5 |
> | **Avg. ↑** | **42.5** | **44.8** | **44.6** | **45.0** |
>
> On the one hand, events in most videos are not sufficiently dense; typically, only one event or shot appears within a short period, so a lower sampling rate may not necessarily miss critical information. On the other hand, as shown above, after **balancing accuracy and computational cost**, we chose `64` in the current paper. We also agree that the issue raised by the reviewer is meaningful and could motivate future work on finer-grained video understanding.

---

> > ### Author Rebuttal · Reviewer_VYQA · 2026-04-03
> >
> > Thanks for the authors' response. I will raise my score to 5. Looking forward to your open-sourced training data.

---

> > > ### Author Response · Authors · 2026-04-03
> > >
> > > Thanks for your ackonwledge of our work and for raising score. Thanks again for your time and effort in reviewing our paper. Your positive feedback is highly encouraging, and we will continue to revise and improve the paper accordingly. We will also release the code and data once accepted!
> > >
> > > Wish you all the best.

---

### Official Review · Reviewer_U5dR · 2026-03-09

**Soundness:** 3
**Presentation:** 3
**Significance:** 3
**Originality:** 3
**Overall Recommendation:** 5
**Confidence:** 4

**Summary:**

This paper studies modality interference in omnimodal large language models, particularly in audio-visual contexts where incorporating audio sometimes degrades visual reasoning performance. The authors propose OmniVideo-R1, a two-stage reinforcement learning (RL) framework to mitigate modality bias. The first stage, Query-intensive Grounding, introduces self-supervised grounding through time-caption pairs with rewards for format, answer quality, consistency, and completeness. The second stage, Modality-attentive Fusion, employs a contrastive reward that encourages superior performance under joint audio-visual input compared to single-modality input. Experiments on multiple audio-visual benchmarks show improvements over strong open- and closed-source baselines.

**Compliance With Llm Reviewing Policy:**

Affirmed.

**Final Justification:**

Thanks for the authors' response, which satisfactorily addressed my concerns. I would like to increase my final rating to a clearer Accept.

**Key Questions For Authors:**

NA

**Limitations:**

yes

**Strengths And Weaknesses:**

## Strengths
- Modality interference is an important problem in omnimodal or multimodal LLMs, and this paper provides an effective solution.
- The query intention and modality attention-based reinforcement learning framework is well motivated. The rewards are well designed.
- The proposed method is validated on multiple widely used benchmarks, and the results demonstrate its superiority.
## Weaknesses
 It is difficult for me to identify significant weaknesses. I only share some questions and suggestions for the authors’ consideration:
- In the Introduction, the authors first describe the phenomenon that incorporating additional audio information in omni LLMs does not surpass visual-only performance, and then explain that this is due to the imbalanced data distribution in omni LLM pretraining. It would be better to clarify that this may not be the only reason. Moreover, providing more discussion or potential strategies on how to better utilize single-modal and multi-modal data during MLLM pretraining would make the paper more insightful.
- The proposed method involves multiple rewards. I am curious about the time or computational cost of reward computation. The $r_{cons}$ measures timestamp-level reward, while $r_{comp}$ computes sentence/video-level reward. An ablation study could better reveal their respective impacts. The specific model used to obtain $r_{comp}$ is not clearly specified, and there is no discussion about evaluator bias or reward quality. In addition, the modality-attentive fusion employs a hyperparameter $\alpha$ for the $r_{attn}$ reward calculation, but the corresponding ablation is missing.
- The experiments focus primarily on the Qwen3-Omni-30B-A3B backbone. There is no validation on other omni LLM backbones.
- Line 234 contains a typo in $s_i$. The Method section involves multiple prompts used for evaluators; it would be better to clearly indicate the corresponding section numbers in the Appendix where these details are provided.

---

> ### Author Rebuttal · Authors · 2026-03-30
>
> We sincerely thank the reviewer for the thoughtful and constructive feedback. We have carefully considered all comments and will revise the paper accordingly.
>
> ### **Q1. Causes of modality interference**
>
> We appreciate the reviewer's insightful comment. In the revised paper, we will describe it as **an important factor**, rather than the sole cause. We will also add the following discussion:
>
> (i) The information density of visual and audio signals is asymmetric, and audio can introduce noise more easily in some tasks.
>
> (ii) The representation granularity of different modality encoders is not fully aligned, which may cause conflicts during joint decoding.
>
> (iii) For visually dominated tasks, additional audio input may sometimes interfere with the reasoning process.
>
> We will also expand the introduction or discussion section to mention multimodal pretraining strategies such as curriculum learning, modality dropout, and modality-balanced sampling, while emphasizing that our post-training framework is complementary to such pretraining improvements.
>
> ### **Q2. Reward cost, ablation, and hyperparameter analysis**
>
> We thank the reviewer for these helpful suggestions and have conducted additional experiments to address them.
>
> **(1) Reward computation cost.** Using `8xH20` training as an example, we deploy the judger on `64xH20`, with `vllm_parallel_size=8`, `global_batch_size=16`, `context_parallel_size=1`, `tensor_model_parallel_size=4`, `expert_model_parallel_size=4`, and `pipeline_model_parallel_size=2`. Under this setup, we measure the average runtime per rollout from rollout start to reward completion:
>
> | Reward | Granularity | Time / Rollout ↓ |
> | --- | --- | :---: |
> | `r_format` | Sequence-level | ~1.62s |
> | `r_cons` | Timestamp-level | ~6.21s |
> | `r_comp` | Sequence-level | ~7.97s |
> | `r_ans` | Sequence-level | ~4.70s |
> | `r_attn` | Sequence-level | ~8.06s |
>
> The cost differences mainly come from the internal logic of each reward. The main bottlenecks are judge-model inference, video clipping/concatenation, and the multi-branch forward pass in the MA stage. For `r_cons`, parallel evaluation over multiple time-caption pairs reduces runtime from ~19.65s to ~6.21s. We also use vLLM-based acceleration to reduce overhead where possible.
>
> **(2) Judge bias and reward quality.** Regarding judge reliability for the specific reward model choice, we have provided additional evidence in our response to **Reviewer@WFnF-P1**. Additional reward ablations are also provided in our response to **Reviewer@WFnF-P2**.
>
> **(3) MA hyperparameter ablation.** We additionally ablated the MA hyperparameter `alpha`:
>
> | `alpha` | OmniVideoBench ↑ | Daily-Omni ↑ |
> | :---: | :---: | :---: |
> | 0.1 | 44.3 | 82.3 |
> | **0.3** | **44.8** | **82.8** |
> | 0.8 | 44.6 | 82.5 |
> | 1.0 | 44.6 | 82.4 |
>
> The results suggest that a moderate margin is more suitable: if it is too small, it may not sufficiently encourage the model to exploit complementary multimodal information; if it is too large, it may weaken the effect of the reward signal.
>
> ### **Q3. Validation on more backbones**
>
> We agree with this suggestion and added experiments on two additional omni backbones, using GSPO or GRPO depending on whether the model architecture is dense or MoE-based:
>
> | Backbone | Method | OmniVideoBench ↑ |
> | --- | --- | :---: |
> | Qwen3-Omni-30B-A3B | Baseline | 37.0 |
> | Qwen3-Omni-30B-A3B | **OmniVideo-R1** | **44.8** |
> | Qwen2.5-Omni-7B | Baseline | 29.3 |
> | Qwen2.5-Omni-7B | **OmniVideo-R1** | **38.7** |
> | MiniCPM-o-2_6 | Baseline | 29.7 |
> | MiniCPM-o-2_6 | **OmniVideo-R1** | **36.2** |
>
> Our method consistently improves performance across all three backbones, which suggests that OmniVideo-R1 is not limited to a specific architecture and may serve as a generally applicable post-training framework.
>
> ### **Q4. Typos and appendix references**
>
> We thank the reviewer for the careful reading. We will correct the typos and explicitly add appendix section references wherever prompts or judge details are discussed in the method section, so that readers can easily locate them.

---

> > ### Author Rebuttal · Reviewer_U5dR · 2026-04-01
> >
> > Thanks for the authors' response, which satisfactorily addressed my concerns. I would like to increase my final rating to a clearer Accept. Good luck.

---

> > > ### Author Response · Authors · 2026-04-02
> > >
> > > Thanks for your time and effort in reviewing our paper. Your feedback has been extremely helpful in improving the quality of our manuscript. We thank for your raising score, and we will continue to revise and refine the manuscript accordingly.
> > >
> > > Wish you all the best.
> > >
> > > Good luck for you, too.

---

### Official Review · Reviewer_6bfV · 2026-03-11

**Soundness:** 3
**Presentation:** 3
**Significance:** 3
**Originality:** 3
**Overall Recommendation:** 5
**Confidence:** 3

**Summary:**

This paper addresses the "modality interference" phenomenon in omnimodal models, where incorporating audio degrades visual performance. The authors propose OmniVideo-R1, a two-stage reinforcement learning post-training framework. The QI stage trains the model to locate query-relevant audio-visual segments via self-supervised time-caption consistency rewards, while the MA stage employs a contrastive learning reward to encourage superior performance under full multimodal input compared to single-modality input. Built on Qwen3-Omni-30B-A3B, the method outperforms open-source SOTA on multiple audio-visual benchmarks without significant visual-only performance degradation.

**Compliance With Llm Reviewing Policy:**

Affirmed.

**Final Justification:**

The author's reply resolved my questions.

**Key Questions For Authors:**

see weakness.

**Limitations:**

yes.

**Strengths And Weaknesses:**

Strengths:

(1) The two-stage design is logically coherent, separating "where to look" from "how to fuse." The QI stage requires no manual process-level annotations, offering good scalability. Ablation studies systematically verify the contribution of each component and provide comparisons across various training paradigms.

(2) Visual-only performance shows no significant degradation, demonstrating that the method preserves the original visual capabilities as much as possible while enhancing multimodal fusion, which aligns with the core objective of addressing modality interference.

Weakness:
Under the full training pipeline (QI -> MA), I did not find an ablation on the data scale of the MA stage. The w/o QI settings in Table 5 skip the QI stage entirely, which does not reflect how varying the MA data size would affect final performance within the complete pipeline.

---

> ### Author Rebuttal · Authors · 2026-03-30
>
> We sincerely thank the reviewer for the thoughtful feedback. We have carefully considered the comment and added the following experiment.
>
> In the full `QI+MA` setting, the complete dataset contains 88173 samples, which we denote as `88K`. The original subset used at the MA stage contains 12887 samples, denoted as `12K`. We vary the amount of MA-stage data for experiment as follows.
>
> | Stage | MA Data Size | OmniVideoBench ↑ | Daily-Omni ↑ |
> | --- | :---: | :---: | :---: |
> | w/o MA | - | 43.6 | 82.0 |
> | QI+MA | 1K | 44.0 | 82.4 |
> | QI+MA | 5K | 44.5 | 82.6 |
> | QI+MA | 12K (original) | 44.8 | 82.8 |
> | QI+MA | 30K | 44.9 | 82.9 |
> | QI+MA | 50K | 45.0 | 83.0 |
> | QI+MA | 88K | 45.0 | 83.1 |
>
> These results suggest two main observations:
>
> (i) The MA stage consistently improves the full pipeline. For example, on OmniVideoBench, adding only 1K MA samples improves performance from **43.6** to **44.0**.
>
> (ii) The performance continues to improve as the MA data scale increases, although the marginal gain gradually diminishes.
>
> We'd be glad to include this experiment in the revised paper to provide more direct evidence for the effectiveness of the two-stage design.

---

> > ### Author Rebuttal · Reviewer_6bfV · 2026-04-01
> >
> > The author's reply resolved my questions.

---

> > > ### Author Response · Authors · 2026-04-02
> > >
> > > Thanks for your acknowledge of our work and for raising score. Your feedback has been extremely helpful in improving the quality of our manuscript, and we will continue to revise and refine the manuscript accordingly!
> > >
> > > Wish you all the best.

---

### Official Review · Reviewer_WFnF · 2026-03-13

**Soundness:** 3
**Presentation:** 3
**Significance:** 3
**Originality:** 3
**Overall Recommendation:** 5
**Confidence:** 3

**Summary:**

This paper presents OmniVideo-R1, a reinforcement-learning-based post-training framework for improving audio-visual reasoning in omnimodal models by addressing modality interference, where adding audio can hurt visual reasoning. Built on Qwen3-Omni-30B-A3B, the method has two main stages: query-intensive grounding, which teaches the model to identify question-relevant temporal audio-video evidence through self-supervised rewards over generated time-caption pairs, and modality-attentive fusion, which encourages stronger performance from combined audio-video input than from either modality alone. The paper also introduces a filtered and balanced audio-visual training pipeline and reports improved results on multiple audio-visual benchmarks while largely preserving visual-only performance.

**Compliance With Llm Reviewing Policy:**

Affirmed.

**Final Justification:**

The rebuttal addressed all questions.

**Key Questions For Authors:**

See weaknesses above

**Limitations:**

The paper includes a brief limitation section, but it focuses mainly on continued reliance on outcome-based ground truth and possible extension to additional modalities. The impact statement is also minimal and could more meaningfully discuss potential societal risks or misuse.

**Strengths And Weaknesses:**

Strengths:
- The paper addresses a meaningful and timely problem in omnimodal modeling and proposes an intuitive two-stage RL framework with clearly defined reward components for query-intensive grounding and modality-attentive fusion.
- The empirical results are broad, including multiple audio-visual benchmarks, visual-only evaluation, training-strategy comparisons, and ablations.
- Instead of rewarding only the final answer, the paper introduces structured outputs with time-caption pairs and adds reward terms for format compliance, time-caption consistency, and completeness of grounded segments.

Weaknesses:
- Key parts of both data filtering and reward computation depend on large external models such as Gemini-2.5-Pro and Qwen3-VL. The paper gives limited analysis of how reliable or stable these judgments are.
- The current ablations show that components matter, but do not analyze sensitivity to reward weights, judge choice, or training stability in much depth.
- The paper combines RL post-training, structured intermediate outputs, grounding-style supervision, and multimodal contrastive comparison in a useful way. The originality comes more from the combination than from a clearly new technical contribution.

---

> ### Author Rebuttal · Authors · 2026-03-30
>
> We sincerely thank the reviewer for the thoughtful and detailed feedback. We have carefully considered all comments and will revise the paper accordingly in the future version.
>
> ### **Q1. Reliability of the external judger**
>
> We agree that the reliability of the external judge is important. Though LLM-as-a-judge is widely used for open-ended evaluation, we still conduct additional analysis to better assess its reliability in our setting, covering both data construction and reward computation.
>
> **(1) Data construction.** We used Gemini-2.5-Pro to judge samples along four dimensions: video dependency, audio dependency, question logic, and response accuracy. To assess the reliability of these judgements, we randomly sampled 100 examples, then manually verified the judgements using a binary pass/fail criterion based on the assigned scores and accompanying rationales from the judger. The judging accuracies are as follows:
>
> | Dimension | Accuracy ↑ |
> | --- | :---: |
> | Video dependency | 1.00 |
> | Audio dependency | 0.98 |
> | Question logic | 0.98 |
> | Response accuracy | 0.97 |
>
> On the same 100 examples, we also judged the 15-category question classification by Qwen-3-32B, which achieved an accuracy of 0.96. Overall, these results suggest that each stage in the data construction pipeline is reasonably reliable, with all measured accuracies no lower than **96%**.
>
> **(2) Reward computation.** Using the 97 response-correct samples from the above subset, covering 485 time-caption pairs, we further evaluated the reward scores produced by Qwen3-VL-235B-A22B-Instruct. We again manually verified the outputs in a binary pass/fail manner based on the reward values and rationales. The resulting accuracies are:
>
> | Reward | Accuracy ↑ |
> | --- | :---: |
> | Consistency | 0.961 |
> | Completeness | 0.94 |
> | Answer | 0.99 |
>
> These results suggest that all reward components are reasonably reliable, with measured accuracies of at least **94%**.
>
> ### **Q2. Reward ablation**
>
> We further ablated the weights of different reward terms. Here, `QI+MA` uses the same QI-stage weights as in the main paper.
>
> | Stage | `r_format` | `r_ans` | `r_intent` | `r_attn` | OmniVideoBench ↑ | Daily-Omni ↑ |
> | --- | :---: | :---: | :---: | :---: | :---: | :---: |
> | **QI** | **1** | **1** | **1** | **-** | **43.6** | **82.0** |
> | QI | 0.5 | 1 | 1 | - | 43.5 | 81.7 |
> | QI | 1 | 0.5 | 1 | - | 42.7 | 81.4 |
> | QI | 1 | 1 | 0.5 | - | 41.0 | 80.3 |
> | QI | 1 | 1 | 0 | - | 38.4 | 75.9 |
> | **QI+MA** | **1** | **1** | **-** | **1** | **44.8** | **82.8** |
> | QI+MA | 0.5 | 1 | - | 1 | 44.6 | 82.5 |
> | QI+MA | 1 | 0.5 | - | 1 | 44.0 | 82.3 |
> | QI+MA | 1 | 1 | - | 0.5 | 43.9 | 82.3 |
> | QI+MA | 1 | 1 | - | 0 | 43.7 | 82.1 |
>
> We also compared the agreement between Qwen3-VL-235B-A22B-Instruct and other judge models using the same protocol as above. Pairwise agreement is reported below:
>
> | Reward Model | Consistency (Cohen's Kappa) ↑ | Completeness (Cohen's Kappa) ↑ | Answer (Cohen's Kappa) ↑ |
> | --- | :---: | :---: | :---: |
> | Gemini-2.5-Pro vs. Qwen3-VL-235B-A22B | 0.78 | 0.82 | 0.97 |
> | GPT-4.1 vs. Qwen3-VL-235B-A22B | 0.84 | 0.86 | 0.98 |
>
> These results suggest that Qwen3-VL-235B-A22B-Instruct serves as a reliable judge, exhibiting strong agreement with other competitive models as indicated by the high Cohen's kappa values, especially with GPT-4.1. They also provide some evidence that our method does not rely on an exceptionally strong omnimodal judger.
>
> ### **Q3. Novelty**
>
> We understand the reviewer's concern and would like to further clarify our motivation here. Rather than simply combining existing components, our method introduces **two coupled designs** for audio-visual post-training. First, **Query Intention (QI)** is not a conventional process-grounding method based on human annotation. It enables self-supervised learning through model-generated time-caption pairs. Second, **Modality Attention (MA)** does not merely reward the final answer. It explicitly compares response quality under unimodal and multimodal conditions, encouraging the model to better exploit cross-modal complementary information rather than simply receiving more inputs.
>
> More importantly, the two stages are **explicitly connected**: QI *learns a reasoning paradigm to establish the basic capability*, while MA learns a fusion strategy that *further improves the model on top of QI*. The `w/o QI` and `w/o MA` results in Tab. 5 of the main text also suggest that either stage alone is weaker than the full two-stage pipeline.
>
> ### **Q4. Limitations**
>
> We appreciate this suggestion and will revise the paper accordingly. In the revised paper, we will expand the broader impacts section. We will more explicitly discuss potential risks, including privacy-sensitive video analysis, cultural biases in the training data, and possible misuse of increasingly capable audio-visual understanding models.

---

> > ### Author Rebuttal · Reviewer_WFnF · 2026-03-31
> >
> > The rebuttal is constructive and addresses judge reliability and reward ablations. The novelty clarification is reasonable, but the technical contribution still appears to arise more from subtle differences over existing ideas than from a fundamentally novel component. Still, the paper makes a meaningful and timely contribution to omnimodal modeling.

---

> > > ### Author Response · Authors · 2026-04-02
> > >
> > > Thank you once again for your valuable comments and constructive suggestions. Your feedback has been extremely helpful in improving the quality of our manuscript. Your positive evaluation is very encouraging, and we will continue to revise and refine the manuscript accordingly!
> > >
> > > Wish you all the best.

---

### Decision · Program_Chairs · 2026-04-30

**Decision:**

Accept (regular)

**Comment:**

This paper proposes OmniVideo-R1, a two-stage reinforcement learning framework that addresses modality interference in omnimodal models by separating query intention identification from modality attention. All four reviewers unanimously recommend acceptance. Reviewer WFnF praised the meaningful and timely problem formulation with broad empirical results. Reviewer 6bfV found the two-stage design logically coherent, separating "where to look" from "how to fuse," with good scalability. Reviewer U5dR stated that it is difficult to identify significant weaknesses and that the method provides an effective solution with well-designed rewards. Reviewer VYQA appreciated the thoughtfully designed two-stage RL strategy and strong performance across benchmarks.

The authors provided comprehensive rebuttals that resolved all remaining concerns. Judge reliability was verified manually at 96%+ accuracy across all dimensions with Cohen's Kappa of 0.78 to 0.98, addressing Reviewer WFnF's concern about reliance on external judge models. A complete MA data scale ablation demonstrated consistent improvement with marginal gains diminishing beyond 12K samples, addressing Reviewer 6bfV. Validation on two additional backbones (Qwen2.5-Omni-7B and MiniCPM-o-2_6) confirmed generalizability, addressing Reviewer U5dR, who raised the score stating "I would like to increase my final rating to a clearer Accept." Direct visual benchmark comparisons showed OmniVideo-R1 surpassing Qwen3-VL on MMStar (73.1 vs. 72.1) and MathVista_mini (80.8 vs. 80.1), and a failure case analysis on 50 OmniVideoBench examples was conducted, addressing Reviewer VYQA, who raised the score from 4 to 5. Reviewer WFnF noted that the novelty arises more from subtle differences over existing ideas than a fundamentally novel component, but maintained the Accept score, confirming the rebuttal addressed all questions.

The paper is technically sound, well-written, and introduces a non-redundant contribution to audio-visual reasoning. The unanimous Accept consensus, thorough experimental validation across multiple backbones and benchmarks, and strong rebuttal responses support acceptance. The paper is useful to the ICML community working on multimodal reasoning and reinforcement learning.